# Deep Homogeneous Mixture Models: Representation, Separation, and Approximation

**Priyank Jaini**
Department of Computer Science & Waterloo AI Institute
University of Waterloo
pjaini@uwaterloo.ca

**Pascal Poupart**
University of Waterloo, Vector Institute & Waterloo AI Institute
ppoupart@uwaterloo.ca

**Yaoliang Yu**
Department of Computer Science & Waterloo AI Institute
University of Waterloo
yaoliang.yu@uwaterloo.ca

## Abstract

At their core, many unsupervised learning models provide a compact representation of homogeneous density mixtures, but their similarities and differences are not always clearly understood. In this work, we formally establish the relationships among latent tree graphical models (including special cases such as hidden Markov models and tensorial mixture models), hierarchical tensor formats and sum-product networks. Based on this connection, we then give a unified treatment of exponential separation in *exact* representation size between deep mixture architectures and shallow ones. In contrast, for *approximate* representation, we show that the conditional gradient algorithm can approximate any homogeneous mixture within $\epsilon$ accuracy by combining $O(1/\epsilon^2)$ "shallow" architectures, where the hidden constant may decrease (exponentially) with respect to the depth. Our experiments on both synthetic and real datasets confirm the benefits of depth in density estimation.

## 1 Introduction

Multivariate density estimation, a widely studied problem in statistics and machine learning [28], is becoming even more relevant nowadays due to the availability of huge amounts of unlabeled data in various applications. Many unsupervised and semi-supervised learning algorithms either implicitly (e.g. generative adversarial networks) or explicitly estimate (some functional of) the underlying density function. In this work, we study the problem of density estimation with an explicit representation through finite mixture models (FMMs) [19], which have endured thorough scientific scrutiny over decades. The popularity of FMMs is largely due to their simplicity, interpretability, and universality, in the sense that, given sufficiently many components (satisfying mild conditions), FMMs can approximate any distribution to an arbitrary level of accuracy [22].

Many familiar unsupervised models in machine learning, at their core, provide a compact representation of homogeneous density mixtures. This list includes (but is not limited to) hidden Markov models (HMM), the recently proposed tensorial mixture models (TMM) [26], latent tree graphical models (LTM)[21], hierarchical tensor formats (HTF) [13], and sum-product networks (SPN) [9; 24]. However, despite all being a certain form of FMM, the precise relationships among these models are not always well-understood. Our first contribution fills this gap: we prove (roughly) that

$\{HMM, TMM\} \subseteq LTM \subseteq HTF \subseteq SPN$. Moreover, converting from a lower to an upper class can be achieved in linear time and without any increase in size. Our results not only clarify the similarities and subtle differences between these widely-used models, but also pave the way for a unified treatment of many properties of such models, using tools from linear algebra.

We next investigate the consequence of converting a deep mixture model into a shallow one. We first prove that the (nonnegative) tensor rank exactly characterizes the minimum size of a shallow SPN (or LTM or HTF due to equivalence) that represents a given homogeneous mixture. Then, we show that a *generic* "deep" SPN (with depth at least 2) can be *exactly* represented by a shallow SPN only when the latter contains exponentially many product nodes. Our result extends significantly those in [7; 26; 10; 18; 8] in various aspects, but most saliently from the restrictive full binary tree [7; 26] to *any* rooted tree. As a consequence, our results imply that a generic HMM (whose underlying tree is "completely" unbalanced) cannot be exactly represented by any polynomially-sized shallow SPN, which, to our best knowledge, has not been shown before.

From a practical point of view, *exact* representations are an overkill: it suffices to *approximate* a given density mixture with reasonable accuracy. Our third contribution demonstrates that under the $\ell_\infty$ metric, we can approximate any homogeneous density mixture within $\epsilon$ accuracy by combining $O(1/\epsilon^2)$ shallow SPNs. However, our proof requires the knowledge of the target density hence is not practical. Instead, borrowing a classic idea from [17] we show that minimizing the KL divergence using the conditional gradient algorithm can also approximate any homogeneous mixture within $\epsilon$ accuracy by combining $O(1/\epsilon^2)$ base SPNs, where the hidden constant decreases exponentially wrt the depth of the base SPNs. Each iteration of the conditional gradient algorithm amounts to learning a base SPN hence can be efficiently implemented. We conduct thorough experiments on both synthetic and real datasets and confirm the benefits of depth in density estimation.

We proceed as follows: In §2 we introduce homogeneous density mixtures. In §3 we articulate the relationships among various popular mixture models. §4 examines the exponential separation in *exact* representation size between deep and shallow models while §5 turns into *approximate* representations. We report our experiments in §6 and finally we conclude in §7. All proofs are deferred to Appendix C.

## 2 Density Estimation using Mixture Models

In this section, we introduce our main problem: how to estimate a multivariate density through an explicit, finite homogeneous mixture. To set up the stage, let $\mathbf{x} = (x_1, \ldots, x_d)$, with $x_i \in \mathbb{X}_i$ where each $\mathbb{X}_i$ is a Borel (measurable) subset of the Euclidean space $\mathbb{E}_i$. We equip a Borel measure $\mu_i$ on $\mathbb{X}_i$. All our subsequent measure-theoretic definitions are w.r.t. the Borel $\sigma$-field of $\mathbb{X}_i$ and the measure $\mu_i$. Let $\mathbb{X} = \mathbb{X}_1 \times \cdots \times \mathbb{X}_d$ and $\mu = \mu_1 \times \cdots \times \mu_d$ be the product space and product measure, respectively. For each $i \in [d] := \{1, \ldots, d\}$, let $\mathcal{F}_i$ be a class of density functions (w.r.t. $\mu_i$) of the variable $x_i$, and let $\mathcal{G}_i = \text{conv}(\mathcal{F}_i)$ be its convex hull. The function class $\mathcal{F}_i$ is essentially our basis of densities for the variable $x_i$. Our setting here follows that in [18] and includes both continuous and discrete distributions.

We are interested in constructing a finite density mixture [19], using component densities from the basis class $\mathcal{F} = \bigcup_{i=1}^{d} \mathcal{F}_i$. We assume that our finite mixture $f$ is "**homogeneous**," i.e.

$$f(\mathbf{x}) = \sum_{j_1=1}^{k_1} \sum_{j_2=1}^{k_2} \cdots \sum_{j_d=1}^{k_d} \mathcal{W}_{j_1, j_2, \ldots, j_d} \prod_{i=1}^{d} f_{j_i}^i(x_i) = \langle \mathcal{W}, \vec{f^1}(x_1) \otimes \cdots \otimes \vec{f^d}(x_d) \rangle, \quad (1)$$

where $\vec{f^i} := (f_1^i, \ldots, f_{k_i}^i) \in \mathcal{F}_i^{k_i}$, $\mathcal{W} \in \bigotimes_i \mathbb{R}_+^{k_i} \simeq \mathbb{R}_+^{k_1 \times \cdots \times k_d}$ is a $d$-order density tensor (nonnegative and sum to 1), and $\langle \cdot, \cdot \rangle$ is the standard inner product on the tensor product space. We refer to the excellent book [13] and Appendix A for some basic definitions about tensors. By dropping linearly dependent densities in each $\mathcal{F}_i$ we can assume w.l.o.g. the tensor representation $\mathcal{W}$ is *unique*.

There are a number of reasons for restricting to homogeneous mixtures: Firstly, this is the most common choice for estimating a multivariate density function [28]. Secondly, we can always apply the usual "homogenization" trick, i.e., by enlarging the function class $\mathcal{F}_i$ and appending the (improper) density 1 to each $\mathcal{F}_i$. Thirdly, homogeneous densities are "universal" if each class $\mathcal{F}_i$ is, c.f. Appendix A of [26]. In other words, any joint density can be approximated arbitrarily well by a homogeneous density, provided that each marginal class $\mathcal{F}_i$ can approximate any marginal density arbitrarily well and the size (i.e. $k_i$) tends to $\infty$. See Appendix F.1 for some empirical verifications, where we show that convex combinations of relatively few isotropic Gaussians can approximate mixtures of

Gaussians of full covariance matrices surprisingly well. Lastly, as we argue below, many known models in machine learning are simply compact representations of homogeneous mixtures.

## 3   Compact Representation of Homogeneous Mixtures

We now recall a few unsupervised learning models in machine learning and show that they have a compact representation of homogeneous mixtures at their core. We prove the precise relationship amongst them. Our results clarify the similarity and difference of these recent developments, and pave the way for a unified treatment of depth separation (Section 4) and model approximation (Section 5).

**Sum-Product Networks (SPN) [9; 24; 18]**   An SPN $\mathfrak{T}$ is a rooted tree whose *leaves are density functions* $f_j^i(x_i)$ over each of the variables $x_1, \ldots, x_d$ and whose internal nodes are either a sum node or a product node. Each edge $(u, v)$ emanating from a sum node $u$ has an associated nonnegative weight $w_{uv}$. The value $\mathfrak{T}_v$ at a product node $v$ is the product of the values of its children, $\prod_{u \in ch(v)} \mathfrak{T}_u$. The value $\mathfrak{T}_u$ at a sum node $u$ is the weighted sum of the values of its children, $\sum_{v \in ch(u)} w_{uv} \mathfrak{T}_v$. The value of an SPN $\mathfrak{T}$ is the expression evaluated at the root node, which we denote as $\mathfrak{T}(\mathbf{x})$. The *scope* of a node $v$ in an SPN is the set of all variables that appear in the leaves of the sub-SPN rooted at $v$. We only consider *decomposable* and *complete* SPNs, i.e., the children of each sum node must have the same scope and the children of each product node must have disjoint scopes. The main advantage of a decomposable and complete SPN over a generic graphical model is that joint, marginal and conditional queries can be answered by two network evaluations and hence, exact inference takes linear time with respect to the size of the network [9; 24; 18]. In comparison, inference in Bayesian Networks and Markov Networks may take exponential time in terms of the size of the network. W.l.o.g. we can rearrange an SPN to have alternating sum and product layers (see Theorem C.1).

The latent variable semantics [23] as well as SPNs representing a mixture model over its leaf densities [24] is well-known. It is also informally known that many tractable graphical models can be treated as SPNs, but precise characterizations are scarce (see [29] which relates SPNs with Bayesian Networks).

**Self-similar SPNs ($\text{S}^3\text{PN}$)**   We call an SPN self-similar, if at *every* sum node, the sub-tree rooted at each of its (product node) children is the same, except the weights at corresponding sum nodes and the densities (but not the variables) at corresponding leaf nodes may differ. This special class of SPNs is exactly equivalent to some recently proposed unsupervised learning models, as we show below.

**Hierarchical Tensor Format (HTF) [13]**   We showed in (1) that a homogeneous mixture can be identified with a tensor $\mathcal{W}$, whose explicit storage can, however, be quite challenging since its size is $\prod_{i=1}^d k_i$. HTF [13] aims at representing tensors compactly, hence can also be used for representing homogeneous mixtures. An HTF consists of a dimension-partition rooted tree (DPT) $\mathsf{T}$, $d$ vector spaces $\mathsf{V}_i$ with bases[1] $\mathcal{F}_i$ at the $d$ leaf nodes, and at most $d-1$ internal nodes which are certain *subspaces* of the tensor product of vector spaces at *disjoint* children nodes. Note that the dimension of the tensor product $\mathsf{U} \otimes \mathsf{V}$ is the product of the dimensions of $\mathsf{U}$ and $\mathsf{V}$. The key in HTF is to truncate each tensor product with a (much smaller) subspace, hence keeping the total storage manageable. Moreover, at each internal node $v$ with $k$ children nodes $\{v_i\}$, instead of storing its $r$ bases directly, we store $r$ coefficient tensors $\{w_{j_1, \ldots, j_k}^{v, \gamma} : \gamma \in [r]\}$ such that, recursively, the $\gamma$-th basis at node $v$ is $\sum_{j_1} \cdots \sum_{j_k} w_{j_1, \ldots, j_k}^{v, \gamma} \mathbf{v}_{j_1} \otimes \cdots \otimes \mathbf{v}_{j_k}$, where $\{\mathbf{v}_{j_i}\}$ consists of the bases at the $i$-th child node $v_i$. To our best knowledge, HTFs have not been recognized as SPNs previously, although they have been used in a spectral method for latent variable models [27].

To turn an HTF into an SPN, more precisely an $\text{S}^3\text{PN}$, we start from the root of the dimension-partition tree $\mathsf{T}$. For each internal node $v$ with say $r$ bases and say $k$ children nodes $\{v_i\}$, each of which has $r_i$ bases themselves, we create three layers in the corresponding $\text{S}^3\text{PN}$: in the first layer we have $r$ sum nodes $\{\mathsf{S}_\gamma^v\}$, each of which is (fully) connected, with respective weights $w_{j_1, \ldots, j_k}^{v, \gamma}$, to the second layer of $\prod_{i=1}^k r_i$ product nodes $\{\mathsf{P}_{j_1, \ldots, j_k}^v\}$, and finally the third layer consists of $\sum_{i=1}^k r_i$ sum nodes $\{\mathsf{S}_{j_i}^{v_i}\}$. The product node $\mathsf{P}_{j_1, \ldots, j_k}^v$ is connected to $k$ sum nodes $\{\mathsf{S}_{j_1}^{v_1}, \ldots, \mathsf{S}_{j_k}^{v_k}\}$. Note that the weights $w_{j_1, \ldots, j_k}^{v, \gamma}$ need not be positive or sum to 1 in HTF, although for representing a homogeneous mixture we can make this choice and we call this subclass $\text{HTF}_+$. Clearly, our construction is reversible hence we can turn an $\text{S}^3\text{PN}$ into an equivalent $\text{HTF}_+$ as well. The construction takes linear time and there is

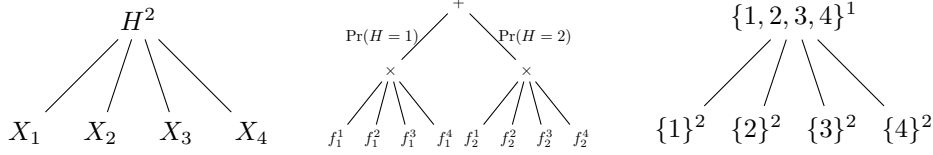

Figure 1: Left: A simple latent class model (special case of LTM). The superscript 2 indicates the number of values the hidden variable $H$ can take. Middle: The equivalent S³PN, where $f_j^i(x_i) = p(X_i = x_i | H = j)$ is from the density class $\mathcal{F}_i$. Right: The dimension-partition tree in an equivalent HTF$_+$. The superscript indicates the number of bases, which should be the same for sibling nodes.

no increase of representation size. See Figs.1,5 for simple illustrations[2]. In summary, HTF *is exactly* S³PN *with arbitrary weights*.

**Diagonal HTF (dHTF) [13]** For later reference, let us call the subclass of HTFs whose coefficient tensors $w_{j_1,\dots,j_k}^{v,\gamma}$ (that define bases recursively at internal nodes of the DPT, see above) are diagonal for all $v$ and $\gamma$ as dHTF, i.e., siblings in the DPT must have the same number of bases ($r_i \equiv r$) and $w_{j_1,\dots,j_k}^{v,\gamma} \neq 0$ only when $j_1 = \dots = j_k$. In neural network terminology, dHTFs are "locally connected." Compared to the fully connected HTF, dHTFs significantly reduce the representation size (at the expense of expressiveness, see Figure 7). For instance, the $\prod_{i=1}^k r_i = r^k$ product nodes in the above conversion from HTF to S³PN are reduced to merely $r$ product nodes.

**Latent Tree Models (LTM) [21; 27; 5]** An LTM is a rooted tree graphical model with observed variables $X_i$ on the leaves and hidden variables $H_j$ on the internal nodes. Note that we allow observed variables $X_i$ to be either continuous or discrete but the hidden variables $H_j$ can take only finitely many values. Using conditional independence, the joint density of observed variables is given as

$$f(x_1, \dots, x_d) = \sum_{h_1} \cdots \sum_{h_t} \mathcal{W}(h_1, \dots, h_t) \prod_{i=1}^d f_{h_{\pi_i}}^i(x_i), \qquad (2)$$

where $H_{\pi_i}$ is the parent of $X_i$. From (2) it is clear that an LTM is a homogeneous density mixture, whose tensor representation is given by the joint density $\mathcal{W}$ of the hidden variables. What is less known[3] is that LTMs are a special subclass of **self-similar** SPNs. It may appear that the size of S³PN is larger than that of an equivalent LTM, but this is because S³PN also encodes the conditional probability tables (CPT) into its structure whereas LTMs require other means to store CPTs. Note also that to evaluate an LTM, one usually needs to run a separately designed algorithm (such as message passing), while in S³PN we evaluate the leaf densities and propagate in linear time to the root. In summary, LTM *is a subclass of* S³PN *with CPTs encoded as edge weights and with inference simplified as network propagation*. More precisely, LTM is exactly dHTF$_+$, since conditioned on the parent, all children nodes must depend on the same realization. An algorithm for converting LTMs into equivalent S³PNs, along with more examples (Figs. 1-6), can be found in Appendix B.1.

**Tensorial Mixture Models (TMM) [26; 7; 6]** TMM [26] is a recently proposed subclass of dHTF$_+$ where nodes on the same *level* of the dimension-partition tree must have the same number of bases. Clearly, TMM is a strict subclass of LTM since the latter only requires sibling nodes in the DPT to have the same number of bases. We note that TMM, as defined in [26], also assumes the DPT to be binary and balanced, i.e. each internal node has exactly two children, although this condition can be easily relaxed. See Figure 2 and its reduced form in Appendix B.3 for a simple example. Further, in Appendix B.4, we give an example of an LTM that is not a TMM.

**Hidden Markov Models (HMM) [3; 25]** HMM is a strict subclass of LTM. [14] recently observed that HMM is equivalent to the tensor-train format, a special subclass of dHTF$_+$ where the DPT is binary and completely "imbalanced." See Appendix B.5 for a simple example. In some sense, TMM and HMM are the two opposite extremes within dHTF$_+$ (or equivalently LTM).

Further, in Appendix B.6 we give an example of an S³PN that is not an LTM, and in Appendix B.7, we give an example of an SPN that is not an S³PN, leading to the following summary:

**Theorem 3.1.** *{TMM, HMM} ⊆ LTM = dHTF$_+$ ⊆ HTF$_+$ = S³PN ⊆ SPN, in the sense that we can convert in linear time from a lower representation class to an upper one, without any increase in size.*

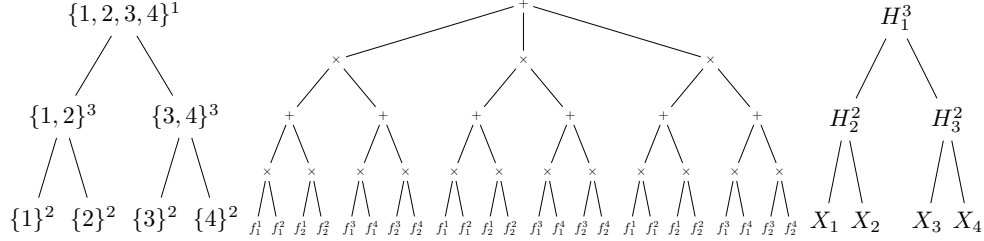

Figure 2: Left: A dimension-partition tree in HTF. The superscripts indicate the number of bases, which should remain constant on each level. Middle: The equivalent S³PN. The leaf $f_j^i$ is the $j$-th basis of vector space $\mathsf{V}_i$. Right: An equivalent TMM. The superscripts indicate the number of values each hidden variable can take (again, remaining constant on each level).

It is important to point out one subtlety here: any (complete and decomposable) SPN, if expanded at the root, is a homogeneous mixture (c.f. (1)). Hence, any SPN is even equivalent to an LCM (i.e. an LTM with one hidden variable taking many values, like in Figure 1), at the expense of potentially increasing the size (significantly). Thus, the containment in Theorem 3.1 should be understood under the premise of not increasing the representation size. It would be interesting to understand if the containment is strict if only polynomial increase in size is allowed. We provide more comparing examples in Appendix B for different models, and in the next section we discuss the (huge) size consequence from converting a certain upper representation class to some lower one.

## 4 Depth Separation

In the previous section, we established relationships among different representation schemes for homogeneous density mixtures. In this section, we prove an exponential separation in size when converting one representation to another and extend the results in [10; 18; 7; 26]. The key is to exploit the equivalence to HTF, which allows us to bound the model size using linear algebra.

We call a (complete and decomposable) SPN shallow if it has only one sum node, followed by a layer of product nodes. Using the equivalence in Section 3, we know a shallow SPN (trivially self-similar) is equivalent to an LCM (a latent tree model with one hidden node taking as many values as the number of product nodes), or an HTF$_+$ whose DPT has depth 1 (c.f. Figure 1). Recall that $\mathrm{rank}_+(\mathcal{W})$ denotes the nonnegative rank of a tensor and $\mathrm{nnz}(\mathcal{W})$ is the number of nonzeros (c.f. Appendix A). The leaf nodes in SPN (LTM) or the leaf bases in HTF are either from $\mathcal{F}$ (union of *linearly independent* component densities) or $\mathcal{G}$ (the convex hull), see the definitions in Section 2.

Our first result characterizes the model capacity of shallow SPNs (LCMs):

**Theorem 4.1.** *If a shallow SPN $\mathfrak{T}$, with leaf (input) nodes from $\mathcal{G}$, represents the density mixture $\mathcal{W}$, then $\mathfrak{T}$ has at least $\mathrm{rank}_+(\mathcal{W})$ many product nodes. Conversely, there always exists a shallow SPN that represents $\mathcal{W}$ using $\mathrm{rank}_+(\mathcal{W})$ product nodes and 1 sum node.*

In other words, the nonnegative rank characterizes the smallest size of shallow SPNs (LCMs) that represent the density mixture $\mathcal{W}$. Similarly, we can prove the following result when the leaf nodes are from $\mathcal{F}$ instead of the convex hull $\mathcal{G}$.

**Theorem 4.2.** *If a shallow SPN $\mathfrak{T}$, with leaf nodes from $\mathcal{F}$, represents the density mixture $\mathcal{W}$, then either $\mathfrak{T}$ has at least $\mathrm{nnz}(\mathcal{W})$ product nodes or $\mathrm{rank}_+(\mathcal{W}) = 1$. Conversely, there always exists a shallow SPN that represents $\mathcal{W}$ using $\mathrm{nnz}(\mathcal{W})$ product nodes and 1 sum node.*

Note that we always have $\mathrm{rank}(\mathcal{W}) \leq \mathrm{rank}_+(\mathcal{W}) \leq \mathrm{nnz}(\mathcal{W})$, thus the lower bound in Theorem 4.2 is stronger than that in Theorem 4.1. This is not surprising, because an SPN with leaf nodes from $\mathcal{G}$ is the same as an SPN with leaf nodes from $\mathcal{F}$ and with an additional layer of sum nodes appended at the bottom (to perform the convex hull operation). This difference already indicates that an additional layer of sum nodes at the bottom can strictly increase the expressive power of SPNs. This distinction between leaf nodes from $\mathcal{F}$ or from $\mathcal{G}$, to our best knowledge, has not been noted before.

The significance of Theorem 4.1 and Theorem 4.2 is that they give *exact* characterizations of the model size of shallow SPNs, and they pave the way for comparing more interesting models. For convenience, we state our next result in terms of LTMs, but the consequence for dHTFs or SPNs should be clear, thanks to the equivalence in Theorem 3.1.

**Theorem 4.3.** *Let an LTM* $\mathsf{T}$ *have $d$ observed variables $\mathcal{X} = \{X_1, \ldots, X_d\}$ with parents $H_i$ taking $r_i$ values respectively. Assuming the CPTs of* $\mathsf{T}$ *are sampled from a continuous distribution, then almost surely, the tensor representation $\mathcal{W}$ for* $\mathsf{T}$ *has rank at least*

$$\max_{1 \leq m \leq d/2} \quad \max_{\{S_1, \ldots, S_m, \bar{S}_1, \ldots, \bar{S}_m\} \subseteq \mathcal{X}} \quad \prod_{i=1}^{m} \min\{r_i, \bar{r}_i, k_i, \bar{k}_i\}, \tag{3}$$

*where $k_i$ ($\bar{k}_i$) is the number of (linearly independent) component densities that $S_i$ ($\bar{S}_i$) has, and $S_i$ ($\bar{S}_i$) are non-siblings.*

**Corollary 4.4.** *In addition to the setting in Theorem 4.3, if each observed variable $X_i$ has $b$ sibling observed variables and $r_i \equiv r \leq k \equiv k_i$, then the tensor representation $\mathcal{W}$ has rank at least $r^{\lfloor d/b \rfloor}$.*

**Corollary 4.5.** *In addition to the setting in Theorem 4.3, if each observed variable $X_i$ has no sibling observed variables and $r_i \equiv r \leq k \equiv k_i$, then the tensor representation $\mathcal{W}$ has rank at least $r^{\lfloor d/2 \rfloor}$.*

Combining Corollary 4.4 with Theorem 4.2 we conclude that an LTM $\mathsf{T}$ with $d$ observed variables $X_i$ where every $b$ of them share the same hidden parent node is equivalent to an LCM $\mathsf{T}'$ where the hidden node must take at least $r^{\lfloor d/b \rfloor}$ many values. Note that $\mathsf{T}$ has $\Theta(d/b)$ hidden variables, each of them taking $r$ values, thus the total size of the CPTs of $\mathsf{T}$ is $\Theta(rd/b)$ while the total size of that of $\mathsf{T}'$ is $r^{\lfloor d/b \rfloor}$, an exponential blow-up. By combining Corollary 4.5 with Theorem 4.2 a similar conclusion can be made for converting an HMM into a LCM. Of course, interpretation using SPNs is also readily available: Almost all depth-$L$ S$^3$PNs ($L \geq 2$) with weights sampled from a continuous distribution can be written as a shallow SPN with necessarily exponentially many product nodes.

To our best knowledge, [10] was the first to construct a polynomial that, while representable by a polynomially-sized depth-$\log d$ SPN, would require exponentially many product nodes if represented by a shallow SPN. However, the deep SPN given in [10, Figure 1] is not complete. Recently, [7] proved that the existence result of [10] is in fact *generic*. However, the results of [7] and subsequent work [26] are limited to full binary trees. In contrast, our general Theorem 4.3 holds for any tree, and we allow non-sibling nodes to take different number of values. As a result, we are able to handle HMMs, the opposite extreme of TMM. Another important point we want to emphasize is that the exponential separation from a shallow (i.e. depth-1) tree can be achieved by increasing the depth by merely 1, as opposed to the depth-$\log d$ constructions in [10; 26].

We end this section by making another observation about Theorem 4.3: It also allows us to compare the model size of LTMs $\mathsf{T}_1$ and $\mathsf{T}_2$ where say $\mathsf{T}_1$, after removing its root $R$, is a subtree of $\mathsf{T}_2$. Indeed, in this case we need only define the children nodes of $R$ as "observed" variables. Then, $\mathsf{T}_1$ becomes an LCM and $\mathsf{T}_2$ serves as $\mathsf{T}$ in Theorem 4.3, with observed variables as the children nodes of $R$. This essentially extends [7, Theorem 3] from a full binary tree to any tree and allowing non-sibling nodes to take different number of values.

## 5 Approximate Representation

In the previous section, we proved that homogeneous mixtures representable by "deep" architectures (such as SPN or LTM) of polynomial size cannot be *exactly* represented by a shallow one with sub-exponential size. In this section, we address a more intricate and relevant question: What if we are only interested in an *approximate* representation?

To formulate the problem, let $g$ and $h$ be two homogeneous mixtures with tensor representation $\mathcal{W}$ and $\mathcal{Z}$, respectively. We consider the distance $\mathrm{dist}(g, h) := \|\mathcal{W} - \mathcal{Z}\|$ for some norm $\| \cdot \|$ specified later. Using the characterization in Theorem 4.1 we formulate our approximation problem as follows. Let $\Delta$ be a perturbation tensor with $\|\Delta\| \leq \epsilon$. What is the minimum value for $\mathrm{rank}_+(\mathcal{W} + \Delta)$, i.e. the size of a shallow SPN? This motivates the following definition adapted from [1]:

$$\epsilon\text{-}\mathrm{rank}_+(\mathcal{W}) = \min\left\{ \mathrm{rank}_+(\mathcal{W} + \Delta) : \|\Delta\| \leq \epsilon \right\} = \min\left\{ \mathrm{rank}_+(\mathcal{Z}) : \|\mathcal{Z} - \mathcal{W}\| \leq \epsilon \right\}. \tag{4}$$

In other words, $\epsilon\text{-}\mathrm{rank}_+$ is precisely the minimum size of a shallow SPN (LCM) that approximates a specified mixture $\mathcal{W}$ with accuracy $\epsilon$. We can similarly define $\epsilon\text{-}\mathrm{rank}$, where we replace the nonnegative rank with the usual rank in (4). Note that the notion of $\epsilon\text{-}\mathrm{rank}$ depends on the norm $\| \cdot \|$.

$\ell_\infty$**-norm** Let the norm in the definition (4) be the usual $\ell_\infty$ norm, and we signify this choice with the notation $\epsilon\text{-}\mathrm{rank}^\infty$. In this setting, we can prove the following nearly-tight bound on the $\epsilon\text{-}\mathrm{rank}$.

**Theorem 5.1.** *Fix $\epsilon > 0$ and tensor $\mathcal{W} \in \mathbb{R}^{k_1 \times \cdots \times k_d}$. Then, for some (small) constant $c > 0$,*

$$\epsilon\text{-rank}^\infty(\mathcal{W}) \le \frac{c\|\mathcal{W}\|_{\mathrm{tr}}}{\epsilon^2}, \tag{5}$$

*where $\|\mathcal{W}\|_{\mathrm{tr}}$ is the tensor trace norm. A similar result holds for $\epsilon\text{-rank}_+^\infty(\mathcal{W})$. The dependence on $\epsilon$ is tight up to a log factor.*

Note that the representative tensor $\mathcal{W}$ for a homogeneous density mixture $f$ is nonnegative and sums to 1, in which case $\|\mathcal{W}\|_{\mathrm{tr}} \le \|\mathcal{W}\|_1 = 1$. Thus, very surprisingly, Theorem 5.1 confirms that any deep SPN (or any LTM or HTF$_+$) can be approximated by some shallow SPN with accuracy $\epsilon$ under the $\ell_\infty$ metric and with at most $c/\epsilon^2$ many product nodes. Of course, this does not contradict with the impossibility results in [7] and [18], because the accuracy $\epsilon$ there is exponentially small.

Theorem 5.1 remains mostly of theoretical interest, though, because (i) a straightforward application of Theorem 5.1 leads to a disappointing bound on the total variational distance between the two homogeneous mixtures $f$ and $g$, due to scaling by the big constant $\prod_i k_i$; (ii) in practical applications we do not have access to $\mathcal{W}$ so the constructive algorithm in our proof does not apply.

**KL divergence** In contrast to the above $\ell_\infty$ approximation, we now give an efficient algorithm to approximate a homogeneous density mixture $h$, using a classic idea of [17]. We propose to estimate $h$ by minimizing the KL divergence over the convex hull[4] of a hypothesis class H:

$$\min_{\mathcal{W}_g \in \mathrm{conv}(\mathsf{H})} \mathsf{KL}(h\|g), \tag{6}$$

where $\mathsf{KL}(h\|g) := \int h(\mathbf{x}) \log \frac{h(\mathbf{x})}{g(\mathbf{x})} \mathrm{d}\mu(\mathbf{x})$, and $\mathcal{W}_g$ is the representative tensor for the mixture $g$. Following [17], we apply the conditional gradient algorithm [12] to solve (6): Given $g_{t-1}$, we find

$$(\eta_t, f_t) \leftarrow \arg\min_{\eta \in [0,1], \mathcal{W}_f \in \mathsf{H}} \mathsf{KL}(h\|(1-\eta)g_{t-1} + \eta f), \qquad g_t \leftarrow (1-\eta_t)g_{t-1} + \eta_t f_t. \tag{7}$$

One can also simply set $\eta_t = \frac{2}{2+t}$, as is common in practice. Note that (7) can be approximately solved based on an iid sample $\mathbf{x}_1, \ldots, \mathbf{x}_n$ hence is practical:

$$\max_{\eta \in [0,1], \mathcal{W}_f \in \mathsf{H}} \sum_{i=1}^n \log[(1-\eta)g_{t-1}(\mathbf{x}_i) + \eta f(\mathbf{x}_i)]. \tag{8}$$

Using basically the same argument as in [17], the above algorithm enjoys the following guarantee:

$$\mathsf{KL}(h\|g_t) \le c_h \delta / t, \tag{9}$$

where $\delta = \sup\{\log \frac{\langle \mathcal{W}, \vec{f}_1 \otimes \cdots \otimes \vec{f}_d \rangle}{\langle \mathcal{Z}, \vec{f}_1 \otimes \cdots \otimes \vec{f}_d \rangle} : \mathcal{W}, \mathcal{Z} \in \mathsf{H}, \mathbf{x} \in \mathbb{X}\}$, and

$$c_h = \min\{p \ge 0 : \mathcal{W}_h = \sum_{i=1}^p \lambda_i \mathcal{W}_i, \mathcal{W}_i \in \mathsf{H}, \boldsymbol{\lambda} \ge 0, \mathbf{1}^\top \boldsymbol{\lambda} = 1\} \tag{10}$$

is essentially the rank of the mixture $h$ (with tensor representation $\mathcal{W}_h$) w.r.t. the class H.

The important conclusion we draw from the above bound (9) is as follows: First, the constant $c_h$ is no larger than $\prod_i k_i$ if H is any of the classes in Theorem 3.1 (since we only consider *finite* homogeneous mixtures $h$). Second, if the target density $h$ is a small number of combinations of densities in H, then $c_h$ is small and we can approximate $h$ using the algorithm (7) efficiently. Third, $c_h$ can be vastly different for different hypothesis classes H, as shown in Section 4. For instance, if $h$ is a generic TMM and H is the shallow class LCM, then $c_h$ is exponential in $d$, whereas if H is the class TMM, then $c_h$ can be as small as 1. There is a trade-off though, since solving (8) for a simpler class (such as LCM) is easier than a deeper one (such as TMM). We will verify this trade-off in our experiments.

## 6 Experiments

We perform experiments on both synthetic and real world data to reinforce our theoretical findings. Firstly, we present experiments on synthetic data to demonstrate the expressive power of an SPN and the algorithm proposed in (7)-(8) which we call SPN-CG. Next, we present two sets of experiments on real world datasets and present results for image classification under missing data.

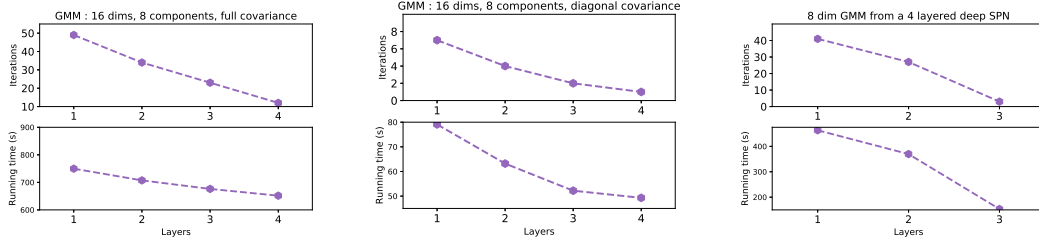

Figure 3: Depth efficiency and performance of SPN-CG

**Synthetic data** Firstly, in appendix F.1 we confirm that a Gaussian mixture model (GMM) with full covariance matrices can be well approximated by a homogeneous mixture model represented by an SPN learned using SPN-CG. Secondly, we generate 20,000 samples from a 16 dimensional GMM under three different settings - (i) 8 component GMM with full covariance matrices, (ii) 8 component GMM with diagonal covariance matrices and, (iii) GMMs represented by a deep SPN with 4 layers - and estimate each using SPN-CG. We consider layers, $L \in \{1, 2, 3, 4\}$ where $L = 1$ corresponds to a shallow network and $L = 4$ corresponds to a network in TMM (a full binary tree). For each $L$, at every iteration of SPN-CG we add a network with $L$ layers. In Figure 3, we plot the number of iterations and the total running time until convergence w.r.t. the depth for each setting described above. We make the following observations: As the depth (layer) increases, the number of iterations decreases sharply, since adding a deeper network effectively is the same as adding exponentially many shallower networks (confirming Section 4). Moreover, although learning a deeper network in each iteration is more expensive than learning a shallower network, the sharp decrease in iterations full compensates this overhead and leads to a much reduced total running time. The advantage in using deeper networks is more pronounced when the data is indeed generated from a deep model.

**Image Classification under Missing Data by Marginalization** A natural setting to test the effectiveness of generative models like deep SPNs is for classification in the regime of missing data. Generative models can cope with missing data naturally through marginalizing the missing values, effectively learning all possible completions for classification. As stated earlier, SPNs are attractive because inference, marginalization and evaluating conditionals is tractable and amounts to one pass through the network. This is in stark contrast with discriminative models that often rely on either data imputation techniques (which result in sub-optimal classification) or by assuming the distribution of missing values is same during train and test time; an assumption that is often not valid in practice.

We perform experiments on MNIST [15] for digit classification and small NORB [16] for 3D object recognition under the MAR (missing at random) regime as described in [26] (Section 3). We experiment with two missing distributions- (i) an i.i.d. mask with a fixed probability of missing each pixel, and (ii) a mask obtained by the union of rectangles of a certain size, each positioned uniformly at random in the image. Concretely, let $P(X, Y)$ be the joint distribution over the images ($X \in \mathbb{R}^d$) and labels $Y \in [M]$. Further, let Z be a random binary vector conditioned on $X = \mathbf{x}$ with distribution $Q(Z|X = \mathbf{x})$. To generate images with missing pixels, we sample $\mathbf{z} \in \{0, 1\}^d$ and consider the vector $\mathbf{x} \odot \mathbf{z}$. A pixel $x_i, i \in [d]$ is considered missing if $z_i = 0$ in which case the corresponding coordinate in $\mathbf{x} \odot \mathbf{z}$ holds $*$ and it holds $x_i$ if $z_i = 1$. In the MAR setting that we consider for our experiments, $Q(Z = \mathbf{z}|X = \mathbf{x})$ is a function of both $\mathbf{z}$ and $\mathbf{x}$ but is independent of changes to $x_i$ if $z_i = 0$ i.e. Z is independent of missing pixels. As described in [26], the optimal classification rule in the MAR regime is $h^*(\mathbf{x} \odot \mathbf{z}) = P(Y = y|w(\mathbf{x}, \mathbf{z}))$ where $w(\mathbf{x}, \mathbf{z})$ is the realization when X coincides with $\mathbf{x}$ on coordinates $i$ for which $z_i = 1$.

Our major goal with these experiments is to test our algorithm SPN-CG for high-dimensional real world settings and show the efficacy of learning SPNs by increasing their expressiveness iteratively. Therefore, we directly adapt the experiments as presented in [26]. Specifically, we adapt the code of HT-TMM for our SPN-CG by following the details in [26]. In each iteration of our algorithm, we add an SPN structure exactly similar to HT-TMM. Therefore, the first iteration of our algorithm (i.e. SPN-CG1) amounts to a structure similar to HT-TMM while additional iterations increase the network capacity. For each iteration, we train the network using an AdamSGD variant with a base learning rate of 0.03 and momentum parameters $\beta_1 = \beta_2 = 0.9$. For each added network structure, we train the model for 22,000 iterations for MNIST and 40,000 for NORB.

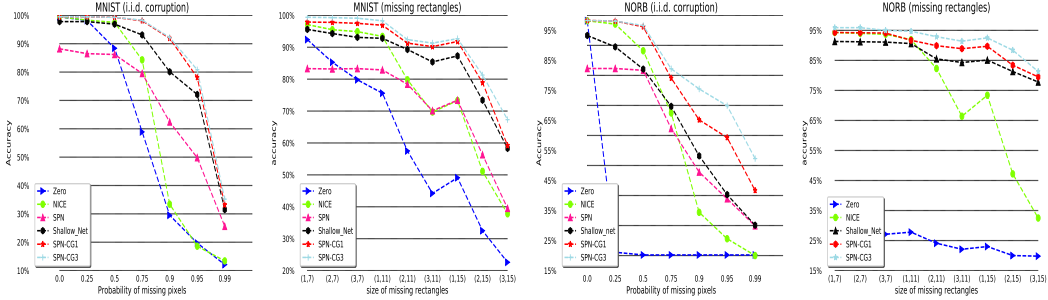

Figure 4: Performance of SPN-CG for missing data on MNIST and NORB

Due to space limit Figure 4 only presents results comparing our model with (i) data imputation techniques that complete missing pixels with zeros or NICE [11], a generative model suited for inpainting, and finally using a ConvNet for prediction, (ii) an SPN with structure learned using data as proposed in [24] augmented with a class variable to maximize joint probability, and (iii) shallow networks to demonstrate the benefits of depth. A more comprehensive figure showing comparisons with several other algorithms is given in appendix F.2, along with details.

SPN-CG1 and SPN-CG3 in Figure 4 stand for one and three iterations of our algorithm respectively. The results show that SPN-CG performs well in all regimes of missing data for both MNIST and NORB. Furthermore, other generative models including SPN with structure learning perform comparably only when a few pixels are missing but perform very poorly as compared to SPN-CG when larger amounts of data is missing. Our results here complement those in [26] where these experiments were first reported with state of the art results.

## 7   Conclusion

We have formally established the relationships among some popular unsupervised learning models, such as latent tree graphical models, hierarchical tensor formats and sum-product networks, based on which we further provided a unified treatment of exponential separation in *exact* representation size between deep architectures and shallow ones. Surprisingly, for *approximate* representation, the conditional gradient algorithm can approximate any homogeneous mixture within accuracy $\epsilon$ by combining $O(1/\epsilon^2)$ shallow models, where the hidden constant may decrease exponentially wrt the depth. Experiments on both synthetic and real datasets confirmed our theoretical findings.

## Acknowledgement

The authors gratefully acknowledge support from the NSERC discovery program.

## Footnotes

[1]More generally frames, in particular, the elements need not be linearly independent.

[2]All of our illustrations of S³PN in the main text are drawn with some redundant leaves, for the sake of making the self-similar property apparent. See Appendix B for the reduced (but equivalent) counterparts.

[3]As an evidence, we note that the recent survey [21] on LTMs did not mention SPNs at all.

[4]This is similar in spirit to [20; 2] which learn mixture of trees, but the algorithms are quite different.

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
