[Supplementary Material]

# Supplementary Material

# Deep Homogeneous Mixture Models:
# Representation, Separation, and Approximation

## A  Tensor Background

For any natural number $d$, we denote $[d] := \{1, \ldots, d\}$. Let $\mathsf{V}_i, i \in [d]$, be $k$-dimensional vector spaces over the real field $\mathbb{R}$, then the tensor product $\mathsf{V}_1 \otimes \cdots \otimes \mathsf{V}_d$ is the canonical vector space that linearizes multilinear maps over the product space $\mathsf{V}_1 \times \cdots \times \mathsf{V}_d$. Perhaps the simplest way to construct the tensor product is to first formally define rank-1 tensors as:

$$\{\mathbf{v}_1 \otimes \cdots \otimes \mathbf{v}_d : \ \mathbf{v}_i \in \mathsf{V}_i, \ i \in [d]\}, \tag{11}$$

and then take the linear span of rank-1 tensors. For each $\mathscr{T} \in \mathsf{V}_1 \otimes \cdots \otimes \mathsf{V}_d$, we define its rank as

$$\mathrm{rank}(\mathscr{T}) := \min\{r : \mathscr{T} = \sum_{\gamma=1}^{r} \mathbf{v}_1^{\gamma} \otimes \cdots \otimes \mathbf{v}_d^{\gamma}, \ \mathbf{v}_i^{\gamma} \in \mathsf{V}_i, \ i \in [d], \ \gamma \in [r]\}. \tag{12}$$

Sometimes we further restrict each factor $\mathbf{v}_i^{\gamma}$ to some subset $\mathsf{U}_i \subseteq \mathsf{V}_i$, leading to a "larger" notion of rank. For instance, when $\mathsf{V}_i \equiv \mathbb{R}^k$, the above definition is called the CP-rank and if we take $\mathsf{U}_i = \mathbb{R}_+^k$, then we get the refined notion of nonnegative rank, denoted as $\mathrm{rank}_+$. Obviously, $\mathrm{rank}_+ \geq \mathrm{rank}$ (whenever the former is defined).

Usually we can identify a $d$-order tensor $\mathscr{T} \in \mathsf{V}_1 \otimes \cdots \otimes \mathsf{V}_d$ with a multi-dimensional array

$$\mathcal{T} = [\mathcal{T}_{i_1, \ldots, i_d}]_{i_j \in [k_i], j \in [d]} \in \bigotimes_i \mathbb{R}^{k_i} \simeq \mathbb{R}^{k_1 \times \cdots \times k_d}, \tag{13}$$

once some bases have been chosen for each $\mathsf{V}_i$. We can extend an inner product to the tensor product space: provided that some inner product $\langle \cdot, \cdot \rangle_i$ has been specified on each $\mathsf{V}_i$, we first define the inner product for rank-1 tensors:

$$\langle \mathbf{u}_1 \otimes \cdots \otimes \mathbf{u}_d, \mathbf{v}_1 \otimes \cdots \otimes \mathbf{v}_d \rangle := \prod_{i=1}^{d} \langle \mathbf{u}_i, \mathbf{v}_i \rangle_i, \tag{14}$$

and then extend multi-linearly.

We give an explicit description of TMM [26] here. For simplicity, let us assume $d = b^L$ for some integers $b$ and $L$. Then, every $d$-order tensor $\mathscr{T}$ can be represented recursively as

$$\phi_{\gamma}^{\ell,t} = \sum_{j=1}^{r_{\ell-1}} w_j^{\ell,t,\gamma} \bigotimes_{s=1}^{b} \phi_j^{\ell-1, b(t-1)+s}, \ \ \ell \in [L-1], \ \gamma \in [r_\ell], \ t \in [b^{L-\ell}], \tag{15}$$

$$\mathscr{T} = \phi_1^{L,1} = \sum_{j=1}^{r_{L-1}} w_j^{L,1,1} \bigotimes_{s=1}^{b} \phi_j^{L-1,s}, \tag{16}$$

where $\phi_{\gamma}^{0,i} \in \mathsf{V}_i$ for all $\gamma \in [r_0]$. Note that the tensor $\mathscr{T}$ is completely determined by

$$\{\phi_{\gamma}^{0,i} : \gamma \in [r_0], i \in [d]\} \bigcup \{\mathbf{w}^{\ell,t,\gamma} \in \mathbb{R}^{r_{\ell-1}} : \ell \in [L-1], \gamma \in [r_\ell], t \in [b^{L-\ell}]\} \cup \{\mathbf{w}^{L,1,1} \in \mathbb{R}^{r_{L-1}}\}, \tag{17}$$

where the former are the base vectors at the bottom level and the latter are the coefficient vectors at each intermediate level. Note that the representation (15)-(16) is not 1-1 (hence some redundancy). Let $\mathsf{TMM}_r^b$ (with default $\mathsf{TMM}_r := \mathsf{TMM}_r^2$) be the class of tensors that can be represented as in (15)-(16).

A simple counting argument reveals that the coefficient tensors in $\mathsf{TMM}_r^b$ have $\frac{d-b}{b-1}r^2 + r$ entries. It is clear that $\mathsf{TMM}_r^b \subseteq \mathsf{TMM}_{r+1}^b$, and $\mathsf{TMM}_1^b$ is exactly the set of rank-1 tensors. As shown in [13], every tensor of rank $r$ can be represented in $\mathsf{TMM}_r^b$. Similarly, every tensor of nonnegative rank $r$ can be represented in $\mathsf{TMM}_r^b$, with all base vectors and coefficient vectors in (17) nonnegative. Moreover, we can normalize the base vectors $\phi_{\gamma}^{0,i}$ so that they have unit $\ell_1$ norm.

Figure 5: Left: A dimension-partition tree in $\text{HTF}_+$. The superscripts indicate the number of bases. Middle: The equivalent $\text{S}^3\text{PN}$. The leaf $f_j^i$ is the $j$-th basis of vector space $\mathsf{V}_i$. Right: An "equivalent LTM." The superscripts indicate the number of values each hidden variable can take. The two densities of $X_3$ are equal, i.e. $f_1^3 = f_2^3$ (hence $X_3$ does not actually depend on $H_1$).

# B  More results on comparing different models

This appendix section provides more details to compliment section 4. We provide additional details and examples to support the arguments that we made in section 4.

## B.1  Converting an LTM to $\text{S}^3\text{PN}$

Given an LTM, we can build a corresponding $\text{S}^3\text{PN}$ as follows: starting from the root of the LTM, for each hidden variable $H$ that takes $k$ possible values $\{1, \ldots, k\}$ and that has $r$ children nodes $\{V_1, \ldots, V_r\}$, we create a sum node $\mathsf{S}_H$ with $k$ children product nodes $\{\mathsf{P}_{H,1}, \ldots \mathsf{P}_{H,k}\}$, each of which has $r$ children sum nodes $\{\mathsf{S}_{V_1}, \ldots, \mathsf{S}_{V_r}\}$. We set the weight from the sum node $\mathsf{S}_H$ to its $i$-th child product node $\mathsf{P}_{H,i}$ as $\Pr(H = i | \pi(H) = j)$, if $\mathsf{S}_H$ connects to the $j$-th child product node of the parent hidden variable $\pi(H)$ (for the root, the parent is empty). If the child $V_t$ is a hidden variable, we continue the construction similarly, while if $V_t = X_i$ is an observed variable, then we replace the sum node $\mathsf{S}_{V_t}$ with the density $f_j^i(x_i)$, assuming $\mathsf{S}_{V_t}$ is connected to the $j$-th child product node of the parent hidden variable $H$. Algorithm 1 summarizes this construction, and Figure 1 illustrates the idea using a simple latent class model (LCM) [21].

In Algorithm 1, we describe a procedure to convert a latent tree model (LTM) as described in (2) to a self-similar SPN ($\text{S}^3\text{PN}$). In Figure 6 we give another example to illustrate Algorithm 1.

Figure 6: Left shows a latent tree model with three binary hidden variables $\mathbf{H} = \{h_1, h_2, h_3\}$ and four observed variables $\mathbf{X} = \{X_1, X_2, X_3, X_4\}$. The second figure shows the equivalent SPN representing the latent tree. The blue edges imply that the hidden variable takes value 0 and the violet edges mean it takes value 1. Only a subset of leaf distributions are explicitly shown in the figure.

In Figure 6, we consider a latent tree graphical model forming a balanced binary tree with three binary hidden variables $\mathbf{H} = \{h_1, h_2, h_3\}$ and four observed variables $\mathbf{X} = \{X_1, X_2, X_3, X_4\}$. The tree has 3 levels and is rooted at $h_1$. The algorithm proceeds by going through each level one at a time. In the first iteration, it encounters the root node $h_1$ and creates a corresponding sum node in the SPN. It then creates two (equal to all possible states of $h_1$) product nodes as children to this sum node.

The edge on the left (blue in the figure) denotes the edge when $h_1 = 0$ and has weight $\Pr(h_1 = 0)$ and the edge on the right (violet) denotes edge when $h_1 = 1$ and has weight $\Pr(h_1 = 1)$. In the next iteration, the algorithm proceeds to level 2 which has two hidden variables $h_2$ and $h_3$. The algorithm processes these one at a time. First, it takes $h_2$ and creates two sum nodes corresponding to $h_2$, one child each for each product node in the previous layer. Next, for two product nodes are created and an edge is created between each of these product nodes and each of the sum node created before corresponding to $h_2$. The same procedure is then repeated for $h_3$. Finally, for each observed variable, a leaf distribution $\Pr(X|\pi_X)$ is induced.

---

**Algorithm 1** Converting an LTM into an S$^3$PN

---

1: **Input :** A latent tree model with $L$ levels and $(\mathbf{X}, \mathbf{H})$
2: **Output :** An equivalent S$^3$PN
3: **for** $l \leftarrow L$ to 1 **do**
4:     $\mathbf{H}_l :=$ {all nodes in current level from left to right order}
5:     **while** $\mathbf{H}_l \neq \emptyset$ **do**
6:         $h = \text{Pop}(\mathbf{H}_l)$
7:         **if** $h \in \mathbf{X}$ **then**
8:             **for** $j \leftarrow 1$ to $|\pi_h|$ **do**
9:                 create a leaf $v_j$ with distribution $Pr(h|\pi_h = j)$
10:                 add an edge $(v_j, p_j^{l-1})$
11:             **end for**
12:         **else**
13:             **if** $\pi_h \neq \emptyset$ **then**
14:                 create $|\pi_h|$ sum nodes i.e. $S_h^l := \{s_1^l, s_2^l, \cdots s_{|\pi_h|}^l\}$
15:                 **for** $j \leftarrow 1$ to $|\pi_h|$ **do**
16:                     add an edge $(s_j^l, p_j^{l-1})$
17:                 **end for**
18:                 create $|h|$ product nodes i.e. $P_h^l := \{p_1^l, p_2^l, \cdots, p_h^l\}$
19:                 **for** $i \leftarrow 1$ to $|\pi_h|$ **do**
20:                     **for** $j \leftarrow 1$ to $|h|$ **do**
21:                         create an edge $(s_i^l, p_j^l)$ with weight $w_{i,j} = Pr(h = j|\pi_h = i)$
22:                     **end for**
23:                 **end for**
24:             **else**
25:                 create one sum node $s_h$
26:                 create $h$ product nodes i.e. $P_h := \{p_1, p_2, \cdots, p_h\}$
27:                 **for** $i \leftarrow 1$ to $|h|$ **do**
28:                     add an edge $(s_h, p_i)$ with weight $Pr(h = i)$
29:                 **end for**
30:             **end if**
31:         **end if**
32:     **end while**
33: **end for**

---

## B.2 dHTF ⊊ HTF

Figure 7 shows the difference between HTF and dHTF. As is clear from the figure, dHTF allows for only local connections and can be thought of as having pointwise multiplication between bases densities. HTF is more general and can allow for cross connections.

(a) HTF

(b) dHTF

Figure 7: Top : A general HTF representation. The network has cross connections and calculates all possible multiplications. Bottom : A dHTF with same bases functions. The representation allows for local connections.

## B.3 Example for TMM as an LTM and S³PN

In fig. 8, we give a representation for fig. 2 without redundancy. The figure shows that a TMM can be represented by an LTM and hence an S³PN.

Figure 8: Left: A dimension-partition tree in HTF. The superscripts indicate the number of bases, which should remain constant on each level. Middle: The equivalent S³PN. The leaf $f_j^i$ is the $j$-th basis of vector space $\mathsf{V}_i$. Right: An equivalent TMM. The superscripts indicate the number of values each hidden variable can take (again, remaining constant on each level).

## B.4 Example for TMM $\subsetneq$ LTM

In fig. 9 we give an S³PN that is equivalent to an LTM but not a TMM. It is evident from the figure that the LTM consists of hidden variables at the same level with different number of possible states. This arrangement, however, is not allowed in TMM.

Figure 9: Left: A dimension-partition tree in HTF$_+$. The superscripts indicate the number of bases. Middle: The equivalent S³PN. The leaf $f_j^i$ is the $j$-th basis of vector space $\mathsf{V}_i$. Right: An equivalent LTM. The superscripts indicate the number of values each hidden variable can take.

A compact representation of fig. 9 without redundancy is given in fig. 10.

Figure 10: Left: A dimension-partition tree in HTF$_+$. The superscripts indicate the number of bases. Middle: The equivalent S³PN. The leaf $f_j^i$ is the $j$-th basis of vector space $\mathsf{V}_i$. Right: An equivalent LTM. The superscripts indicate the number of values each hidden variable can take.

## B.5 TTM/HMM as LTM and S³PN

In fig. 11, we give an example of a Tensor Train Model (TTM) which is know to be equivalent to an HMM. We show an equivalent representation of the TTM/HMM into an LTM and therefore an S³PN.

Figure 11: Left: A dimension-partition tree in tensor-train. The superscripts indicate the number of bases, which should remain constant for siblings. Middle: The equivalent S³PN. The leaf $f_j^i$ is the $j$-th basis of vector space $V_i$. Right: An equivalent HMM. The superscripts indicate the number of values each hidden variable can take.

Figure 12 shows a simpler example to convert a TTM/HMM to an LTM and S³PN with no redundancy.

Figure 12: Left: A dimension-partition tree in HTF₊. The superscripts indicate the number of bases. Middle: The equivalent S³PN. The leaf $f_j^i$ is the $j$-th basis of vector space $V_i$. Right: An "equivalent LTM." The superscripts indicate the number of values each hidden variable can take. The two densities of $X_3$ are equal, i.e. $f_1^3 = f_2^3$ (hence $X_3$ does not actually depend on $H_1$).

## B.6    Example for LTM $\subsetneq$ S$^3$PN

In **??** we give an example of an S$^3$PN whose resulting latent model has cycles and hence cannot be represented as an LTM without increasing the size of the Latent tree exponentially w.r.t. to the size of the latent model.

## B.7    Example for S$^3$PN $\subsetneq$ SPN

In fig. 13, we give an example of an SPN that is not equivalent to S$^3$PN. It is evident from the example that the subtrees rooted at the first sum node have different variable partitions and hence the resulting SPN is not sn S$^3$PN. The figure on the right shows that converting this SPN to an S$^3$PN will result in an increase in the size of the network.

Figure 13: Left: An SPN but which is not an S$^3$PN. The leaf $f_j^i$ is the $j$-th basis of vector space $\mathsf{V}_i$. Right: The equivalent S$^3$PN requires an increase in the size of the network.

## C    Proofs

**Theorem C.1.** *Any SPN can be rearranged to have alternating layers of sum and product nodes without any change in the size of the resultant standard SPN from the original SPN.*

*Proof.* It is straightforward to show that consecutive combination of either sum nodes or product nodes can be merged/collapsed into one layer of the corresponding nodes. This can be seen as follows: consider a sum node $v$ that has $m$ sum nodes as children and denote the set as $ch(v) := \{v_i\}_{i=1}^m$. Then, the expression $f_v$ evaluated at $v$ is

$$f_v(\mathbf{x}) = \sum_{i=1}^m \alpha_{v_i} f_{v_i}(\mathbf{x}) \tag{18}$$

However, since each $v_i \; \forall i \in [m]$ is also a sum node; denote the children of $v_i$ by the set $ch(v_i) := \{\hat{v}_{i,j}\}_{j=1}^{t_i}$ for each $i \in [m]$. Thus,

$$f_{v_i}(\mathbf{x}) = \sum_{j=1}^{t_i} \beta_{\hat{v}_{i,j}} f_{\hat{v}_{i,j}} \tag{19}$$

Therefore, $f_v(\mathbf{x})$ can be now be re-written as

$$f_v(\mathbf{x}) = \sum_{i=1}^m \alpha_{v_i} \sum_{j=1}^{t_i} \beta_{\hat{v}_{i,j}} f_{\hat{v}_{i,j}} \tag{20}$$

$$= \sum_{i=1}^m \sum_{j=1}^{t_i} \alpha_{v_i} \beta_{\hat{v}_{i,j}} f_{\hat{v}_{i,j}} \tag{21}$$

$$\tag{22}$$

Define a $1 - 1$ mapping between the tuple $(i, j)$ $i \in [m]$, $j \in [t_i]$ and $[K]$ where $K = \sum_{i=1}^{m} t_i$ such that $k = j + \sum_{l=1}^{i-1} t_l$, $k \in [K]$. Then, we can re-write the above as

$$f_v(\mathbf{x}) = \sum_{k=1}^{K} \gamma_k f_{\hat{v}_K} \tag{23}$$

where $\gamma_k = \alpha_{v_i} \beta_{\hat{v}_{i,j}}$ and $f_{\hat{v}_k} = f_{\hat{v}_{i,j}}$. This shows that two consecutive layers of sum node can be collapsed into one layer of sum layer while preserving the same size of the network. Similarly, it can be shown for consecutive layers of product nodes.

Now, we give the procedure to convert any SPN into an SPN with alternating layers of sums and products. Perform a top-down pass starting at the root node (W.l.o.g. assume the root node is a sum node). For every children of the root node, if it is a sum node, merge the node into the root node. This ensures that after this step the top layer and the next layer are alternating (including leaf nodes). Proceeding similarly for every node in the network ensures the final network has alternating layers throughout. This completes the proof. □

**Theorem 4.1.** *If a shallow SPN $\mathfrak{T}$, with leaf (input) nodes from $\mathcal{G}$, represents the density mixture $\mathcal{W}$, then $\mathfrak{T}$ has at least $\mathrm{rank}_+(\mathcal{W})$ many product nodes. Conversely, there always exists a shallow SPN that represents $\mathcal{W}$ using $\mathrm{rank}_+(\mathcal{W})$ product nodes and 1 sum node.*

*Proof.* Suppose the shallow SPN $\mathfrak{T}$ represents the (homogeneous) mixture density $\mathcal{W}$. If the hidden layer is all sum nodes, then the output node must be a product node. The claim trivially holds in this case. If the hidden layer is $r$ product nodes, then the output node is a sum node, with weight $z_\gamma$ to the $\gamma$-th product node. The output of the SPN $\mathfrak{T}$, when expanded at the root, is in the following form:

$$\mathfrak{T}(\mathbf{x}) = \sum_{\gamma=1}^{r} z_\gamma \prod_{i=1}^{d} g_i^\gamma(x_i) = \sum_{\gamma=1}^{r} z_\gamma \langle \mathbf{w}_1^{(\gamma)} \otimes \cdots \otimes \mathbf{w}_d^{(\gamma)}, \vec{f}_1(x_1) \otimes \cdots \otimes \vec{f}_d(x_d) \rangle \tag{24}$$

$$= \langle \sum_{\gamma=1}^{r} z_\gamma \mathbf{w}_1^{(\gamma)} \otimes \cdots \otimes \mathbf{w}_d^{(\gamma)}, \vec{f}_1(x_1) \otimes \cdots \otimes \vec{f}_d(x_d) \rangle \tag{25}$$

$$= \langle \mathcal{W}, \vec{f}_1(x_1) \otimes \cdots \otimes \vec{f}_d(x_d) \rangle. \tag{26}$$

Thus, $\mathcal{W} = \sum_{\gamma=1}^{r} z_\gamma \cdot \mathbf{w}_1^{(\gamma)} \otimes \cdots \otimes \mathbf{w}_d^{(\gamma)}$, i.e., $\mathrm{rank}_+(\mathcal{W}) \leq r$.

Conversely, let $r = \mathrm{rank}_+(\mathcal{W})$ with the decomposition $\mathcal{W} = \sum_{\gamma=1}^{r} \mathbf{w}_1^{(\gamma)} \otimes \cdots \otimes \mathbf{w}_d^{(\gamma)}$. Note that each $\mathbf{w}_i^{(\gamma)}$ is nonzero, as otherwise we would be able to reduce the rank. We construct a shallow SPN $\mathfrak{T}$ to represent $\mathcal{W}$ as follows: On the first layer we have $r$ product nodes, with the $\gamma$-th one computing $\prod_{i=1}^{d} g_i^{(\gamma)}(x_i)$, where

$$g_i^{(\gamma)}(x_i) = \sum_{j=1}^{k} \bar{w}_{ij}^{(\gamma)} f_{i,j}(x_i), \quad \bar{w}_{ij}^{(\gamma)} = \frac{w_{ij}^{(\gamma)}}{\|\mathbf{w}_i^{(\gamma)}\|_1}. \tag{27}$$

Note that $\|\mathbf{w}_i^{(\gamma)}\|_1 := \sum_{j=1}^{k} w_{ij}^{(\gamma)} > 0$ hence the above is well-defined. Then, we add a sum node on top of all product nodes, with weight $\|\mathbf{w}^{(\gamma)}\|_1 := \prod_{i=1}^{d} \|\mathbf{w}_i^{(\gamma)}\|_1 > 0$ for the $\gamma$-th product node. The output of the constructed shallow SPN is:

$$f(\mathbf{x}) = \sum_{\gamma=1}^{r} \|\mathbf{w}^{(\gamma)}\|_1 \prod_{i=1}^{d} g_i^{(\gamma)}(x_i) = \sum_{\gamma=1}^{r} \prod_{i=1}^{d} \sum_{j=1}^{k} w_{ij}^{(\gamma)} f_{i,j}(x_i) \tag{28}$$

$$= \sum_{\gamma=1}^{r} \langle \mathbf{w}_1^{(\gamma)} \otimes \cdots \otimes \mathbf{w}_d^{(\gamma)}, \vec{f}_1(x_1) \otimes \cdots \otimes \vec{f}_d(x_d) \rangle = \langle \mathcal{W}, \vec{f}_1(x_1) \otimes \cdots \otimes \vec{f}_d(x_d) \rangle. \tag{29}$$

The proof is now complete. □

**Theorem 4.2.** *If a shallow SPN $\mathfrak{T}$, with leaf nodes from $\mathcal{F}$, represents the density mixture $\mathcal{W}$, then either $\mathfrak{T}$ has at least $\mathrm{nnz}(\mathcal{W})$ product nodes or $\mathrm{rank}_+(\mathcal{W}) = 1$. Conversely, there always exists a shallow SPN that represents $\mathcal{W}$ using $\mathrm{nnz}(\mathcal{W})$ product nodes and 1 sum node.*

*Proof.* Suppose $\mathfrak{T}$ has a hidden layer of sum nodes. Because $\mathfrak{T}$ is standard, the product output is then a mixture density of the following form:

$$\mathfrak{T}(\mathbf{x}) = \prod_{i=1}^{d} \sum_{j=1}^{k} w_{ij} f_{i,j}(x_i) = \langle \mathbf{w}_1 \otimes \cdots \otimes \mathbf{w}_d, \vec{f_1} \otimes \cdots \otimes \vec{f_d} \rangle = \langle \mathcal{W}, \vec{f_1} \otimes \cdots \otimes \vec{f_d} \rangle. \quad (30)$$

Thus, $\mathcal{W} = \mathbf{w}_1 \otimes \cdots \otimes \mathbf{w}_d$ has nonnegative rank 1. On the other hand, if $\mathfrak{T}$ has a hidden layer of product nodes, then the output of the standard SPN $\mathfrak{T}$, when expanded at the root sum node, is in the following form:

$$\mathfrak{T}(\mathbf{x}) = \sum_{j_1=1}^{k} \cdots \sum_{j_d=1}^{k} z_{j_1,\ldots,j_d} \prod_{i=1}^{d} f_{i,j_i}(x_i) = \langle \mathcal{W}, \vec{f_1}(x_1) \otimes \cdots \otimes \vec{f_d}(x_d) \rangle. \quad (31)$$

Thus, $\mathcal{W} = \mathcal{Z}$, in particular $\mathsf{nnz}(\mathcal{W}) = \mathsf{nnz}(\mathcal{Z})$, but the latter is exactly the number of product nodes in $\mathfrak{T}$.

The converse part follows by reversing the above argument. $\qquad\square$

**Theorem 4.3.** *Let an LTM* $\mathsf{T}$ *have $d$ observed variables $\mathcal{X} = \{X_1, \ldots, X_d\}$ with parents $H_i$ taking $r_i$ values respectively. Assuming the CPTs of* $\mathsf{T}$ *are sampled from a continuous distribution, then almost surely, the tensor representation $\mathcal{W}$ for* $\mathsf{T}$ *has rank at least*

$$\max_{1 \leq m \leq d/2} \quad \max_{\{S_1,\ldots,S_m,\bar{S}_1,\ldots,\bar{S}_m\} \subseteq \mathcal{X}} \prod_{i=1}^{m} \min\{r_i, \bar{r}_i, k_i, \bar{k}_i\}, \quad (3)$$

*where $k_i$ ($\bar{k}_i$) is the number of (linearly independent) component densities that $S_i$ ($\bar{S}_i$) has, and $S_i$ ($\bar{S}_i$) are non-siblings.*

*Proof.* We present our proof using the equivalence between LTM and dHTF.

Recall that an HTF consists of a dimension-partition tree (DPT) $\mathsf{T}$ whose leaf nodes $\{i\}, i \in [d]$ represent $d$ vector spaces with bases $\{\mathbf{v}_1^i, \ldots, \mathbf{v}_{r_i}^i\}$ respectively. At each internal node $\beta$ with $b_\beta$ children nodes $\beta_1, \ldots, \beta_{b_\beta}$, we have $r_\beta$ coefficient tensors $\mathbf{w}^{\beta,\ell[\beta]} \in \mathbb{R}^{r_{\beta_1} \times \cdots \times r_{\beta_{b_\beta}}}$, $\ell[\beta] \in [r_\beta]$, and $r_\alpha$ denotes the number of bases at node $\alpha$. Any tensor $\mathcal{W}$ living in the space at the root $D$ of $\mathsf{T}$ can thus be represented using $r_D$ coefficients $\{c_{\ell[D]} : \ell[D] \in [r_D]\}$ in the following way (c.f. Eq (11.26) of [13] for the special case when the DPT is binary):

$$\mathcal{W} = \sum_{\substack{\ell[i]=1 \\ i \in [d]}}^{r_i} \left[ \sum_{\substack{\ell[\alpha]=1 \\ \alpha \in \mathsf{T}\backslash\mathsf{L}}}^{r_\alpha} c_{\ell[D]} \prod_{\beta \in \mathsf{T}\backslash\mathsf{L}} w_{\ell[\beta_1],\ldots,\ell[\beta_{b_\beta}]}^{\beta,\ell[\beta]} \right] \bigotimes_{i=1}^{d} \mathbf{v}_{\ell[i]}^i. \quad (32)$$

For a dHTF, the coefficient tensors $w_{\ell[\beta_1],\ldots,\ell[\beta_{b_\beta}]}^{\beta,\ell[\beta]}$ are diagonal, so in the summation above we can only consider sibling nodes once as a group. The key observation is that the right-hand side of (32) is a sum of many rank-1 tensors, hence $\mathcal{W}$ is likely to have a large rank.

Let $\{S_i, \bar{S}_i : i = 1, \ldots, m\} \subseteq \mathcal{X} := \{X_1, \ldots, X_d\}$, where $S_i$'s are non-siblings and $\bar{S}_i$'s are also non-siblings. Set $t_i = \min\{r_i \bar{r}_i, k_i, \bar{k}_i\}$. For each $S_i$, set its parent say $H_i$'s (diagonal) coefficient tensor as follows:

$$w_{\ell[S_i]}^{H_i,\ell[H_i]} = \begin{cases} 1, & \text{if } \ell[H_i] = \ell[S_i] = \ell[\bar{S}_i] \leq t_i \\ 0, & \text{otherwise} \end{cases}. \quad (33)$$

Similarly for each $\bar{S}_i$. For any remaining internal node $\beta$, set its (diagonal) coefficient tensor as:

$$w_{\ell[\beta_1]}^{\beta,\ell[\beta]} = \begin{cases} 1, & \text{if } \ell[\beta] = 1 \\ 0, & \text{otherwise} \end{cases}. \quad (34)$$

Under the above specification, we have

$$\mathcal{W} \propto \sum_{j_1=1}^{t_1} \cdots \sum_{j_m=1}^{t_m} \underbrace{\left[\otimes_{i=1}^{m} \mathbf{v}_{j_i}^{S_i}\right]}_{\mathbf{a}_{j_1,\ldots,j_m}} \bigotimes \underbrace{\left[\otimes_{i=1}^{m} \mathbf{v}_{j_i}^{\bar{S}_i}\right]}_{\mathbf{b}_{j_1,\ldots,j_m}}. \quad (35)$$

Since $\{\mathbf{a}_{j_1,\ldots,j_m}\}$ and $\{\mathbf{b}_{j_1,\ldots,j_m}\}$ are linearly independent, respectively, through matricization we know $\mathrm{rank}(\mathcal{W}) \geq \prod_i t_i$. This shows that there exist coefficient tensors $w$ so that $\mathrm{rank}(\mathcal{W}) \geq \prod_i t_i$.

To extend our conclusion from a special realization above to the generic case, let us note that the determinant of any submatrix of any matricization of $\mathcal{W}$ is a polynomial function of the coefficient tensors $w$. We have shown above this polynomial is not always zero, but then it follows immediately that the zero set of this polynomial has measure zero [4], i.e., for a generic realization of the coefficient tensors, we have $\mathrm{rank}(\mathcal{W}) \geq \prod_i t_i$. $\qquad\square$

We constructed an explicit homogeneous mixture in the above proof whose rank is exponential. A similar construction, in the discrete setting, is given below [18]:

**Example C.2.** *Let $x_i \in \{0,1\} \forall i$, $k = 2$ and $d = 2m$. Choose for all $\forall i$ the basis (unnormalized) densities $f_{i,1}(x_i) = 1(x_i = 1)$ and $f_{i,2}(x_i) = 1(x_i = 0)$ (wrt some non-degenerate counting measure on $\{0,1\}$). Consider the (unnormalized) multivariate density $F$ on $\{0,1\}^d$ (wrt the product counting measure):*

$$F(\mathbf{x}) = \begin{cases} 1, & \text{if } x_i = x_{i+m} \text{ for all } 1 \leq i \leq m \\ 0, & \text{otherwise} \end{cases}. \tag{36}$$

*Clearly, $F$ is a density mixture with input nodes from $\mathcal{F}$ and the associated tensor $\mathcal{W}$ satisfies $\mathrm{rank}_+(\mathcal{W}) > 1$. Hence, a standard shallow SPN needs at least $\mathrm{nnz}(\mathcal{W}) = 2^{d/2}$ product nodes to represent $F$, with input nodes from $\mathcal{F}$. The density mixture $F$ is the so-called $\mathrm{EQUAL}$ function in [18], whose Theorem 24 follows immediately from our Theorem 4.2 since $\mathrm{nnz}(\mathcal{W}) \geq \mathrm{rank}(W)$ for any matricization $W$ of $\mathcal{W}$.*

*We note that $F$ is also a density mixture with input nodes from $\mathcal{G}$ (elements of which are themselves mixtures of $f_{i,1}$ and $f_{i,2}$), and $\mathrm{rank}_+(\mathcal{W}) \geq \mathrm{rank}(W) = \mathrm{nnz}(\mathcal{W}) = 2^{d/2} \geq \mathrm{rank}_+(\mathcal{W})$. Thus, any standard shallow SPN with input nodes from $\mathcal{G}$ still requires $2^{d/2}$ product nodes to represent the $\mathrm{EQUAL}$ function. In other words, an SPN with an input layer from $\mathcal{F}$, a layer of sum nodes, a layer of product nodes, and a single sum node as output, would still require $2^{d/2}$ product nodes in order to represent the $\mathrm{EQUAL}$ function. This distinction between input nodes from $\mathcal{F}$ and input nodes from $\mathcal{G}$, to our best knowledge, has not been noted before.*

**Theorem 5.1.** *Fix $\epsilon > 0$ and tensor $\mathcal{W} \in \mathbb{R}^{k_1 \times \cdots \times k_d}$. Then, for some (small) constant $c > 0$,*

$$\epsilon\text{-rank}^\infty(\mathcal{W}) \leq \frac{c\|\mathcal{W}\|_{\mathrm{tr}}}{\epsilon^2}, \tag{5}$$

*where $\|\mathcal{W}\|_{\mathrm{tr}}$ is the tensor trace norm. A similar result holds for $\epsilon\text{-rank}_+^\infty(\mathcal{W})$. The dependence on $\epsilon$ is tight up to a log factor.*

*Proof.* Note that the $\ell_\infty$ norm is dominated by the $\ell_2$ norm, so $\epsilon\text{-rank}^2 \geq \epsilon\text{-rank}^\infty$. Thus, given $\mathcal{W}$, we consider the approximation problem:

$$\min_{\|\mathcal{Z}\|_{\mathrm{tr}} \leq \|\mathcal{W}\|_{\mathrm{tr}}} \|\mathcal{Z} - \mathcal{W}\|_2^2. \tag{37}$$

Obviously, the minimum is $0$. Moreover, if we run the generalized conditional gradient of [12] with initialization $\mathcal{Z}_0 = \mathbf{0}$, then after $t$ iterations, we have

$$\|\mathcal{Z}_t - \mathcal{W}\|_2^2 \leq \frac{c\|\mathcal{W}\|_{\mathrm{tr}}}{t}, \;\; \mathrm{rank}(\mathcal{Z}_t) \leq t, \tag{38}$$

where $c$ is some small universal constant. Here we are exploiting the property that each iteration of the conditional gradient algorithm only increments the rank by at most 1. Setting $c\|\mathcal{W}\|_{\mathrm{tr}}/t = \epsilon^2$ gives us

$$\|\mathcal{Z}_t - \mathcal{W}\|_\infty \leq \|\mathcal{Z}_t - \mathcal{W}\|_2 \leq \epsilon, \;\; \mathrm{rank}(\mathcal{Z}_t) \leq \frac{c\|\mathcal{W}\|_{\mathrm{tr}}}{\epsilon^2}, \tag{39}$$

whence follows $\epsilon\text{-rank}^\infty(\mathcal{W}) \leq O(\|\mathcal{W}\|_{\mathrm{tr}}/\epsilon^2)$.

The proof for the nonnegative rank is completely similar. $\qquad\square$

The inverse-square dependence on $\epsilon$ in Theorem 5.1 is almost tight, as shown below:

**Theorem C.3** ([1, Theorem 2.1]). *Let W be a $k^{d/2} \times k^{d/2}$ matrix with $|w_{i,i}| = 1 \ \forall i$ and $|w_{i,j}| \leq \epsilon \ \forall i \neq j$, where $k^{-d/4} \leq \epsilon \leq \frac{1}{2}$. Then, for some absolute positive constant c,*

$$\operatorname{rank}(W) \geq \frac{cd \log k}{\epsilon^2 \log(\frac{1}{\epsilon})} \tag{40}$$

The above theorem, through un-matricization, clearly implies that there exist tensors $\mathcal{W}$ with $\epsilon$-rank$^\infty(\mathcal{W})$ lower bounded by $\frac{cd \log k}{\epsilon^2 \log(\frac{1}{\epsilon})}$, when $\epsilon$ is not too small.

A few remarks with regard to Theorem 5.1 are in order. We note first that our proof actually gives the same upper bound for the epsilon-rank under any $\ell_p$ norm where $p \in [2, \infty]$. Using the norm inequality $\|\mathcal{W}\|_1 \leq k^{d/2}\|\mathcal{W}\|_2$, we then immediately have from Theorem 5.1 that

$$\epsilon\text{-rank}^1(\mathcal{W}) \leq \frac{c\|\mathcal{W}\|_{\mathrm{tr}} k^d}{\epsilon^2}. \tag{41}$$

Note however that there is still a factor of $k^{d/4}$ gap between the upper and lower bounds in Theorem D.2. It might be possible to optimize the lower bound in Theorem D.2 through different matricizations.

There are at least two issues with Theorem 5.1. First, if we use it to naively bound the $L_1$ norm difference of the underlying densities, i.e.,

$$\|g - h\|_1 = \int |\langle \mathcal{W} - \mathcal{Z}, \vec{f}_1(x_1) \otimes \cdots \otimes \vec{f}_d(x_d)\rangle|\mathrm{d}\mu(\mathbf{x}) \leq k^d\|\mathcal{W} - \mathcal{Z}\|_\infty, \tag{42}$$

the big constant $k^d$ would pop out in the worse case. More disturbingly, in a practical application, we usually do not have access to $\mathcal{W}$ (which is too large to maintain directly anyways), hence it is not clear how to attain the bound in Theorem 5.1 algorithmically.

# D  $\ell_1$ **norm based** $\epsilon$-rank

In this section, we fix the norm $\|\cdot\|$ to be the usual $\ell_1$ norm in definition (4), and we use the notation $\epsilon$-rank$^1_+$ or $\epsilon$-rank$^1$ to signify this convention. Our next result provides a new lower bound on the $\epsilon$-rank, based on matricization.

**Theorem D.1.** *Fix $\epsilon > 0$ and let $\mathcal{W} \in \mathbb{R}^{k_1 \times \cdots \times k_d}$. Then,*

$$\epsilon\text{-rank}^1_+(\mathcal{W}) \geq \epsilon\text{-rank}^1(\mathcal{W}) \geq \min\{i \geq 0 : \epsilon \geq \|W\|_{\mathrm{tr}} - \sum_{j=1}^{i} \sigma_i(W)\}, \tag{43}$$

*where $W$ is any matricization of the tensor $\mathcal{W}$, $\|\cdot\|_{\mathrm{tr}}$ is the matrix trace norm (i.e. sum of singular values), and $\sigma_i(W)$ denotes the $i$-th largest singular value of $W$.*

*Proof.* Since the nonnegative rank is lower bounded by the rank, which is in turn lower bounded by the rank of any matricization, we clearly have

$$\epsilon\text{-rank}^1_+(\mathcal{W}) \geq \epsilon\text{-rank}^1(\mathcal{W}) \geq \epsilon\text{-rank}^1(W), \tag{44}$$

where $W$ is an arbitrary matricization of $\mathcal{W}$ (note that matricization does not change the $\ell_1$ norm). Moreover, for matrices, $\|\cdot\|_\infty \leq \|\cdot\|_{\mathrm{sp}}$ (i.e., maximum singular value) hence $\|\cdot\|_1 \geq \|\cdot\|_{\mathrm{tr}}$, thanks to duality. Therefore,

$$\epsilon\text{-rank}^1(W) = \min_{\|\Delta\|_1 \leq \epsilon} \operatorname{rank}(W + \Delta) \geq \min_{\|\Delta\|_{\mathrm{tr}} \leq \epsilon} \operatorname{rank}(W + \Delta) = \min_{\|W - Z\|_{\mathrm{tr}} \leq \epsilon} \operatorname{rank}(Z). \tag{45}$$

Using say [47, Theorem 1], we know that at the minimum we can choose $Z$ to have the same singular vectors as $W$. It is clear then that $Z$ should match the singular values of $W$, from the biggest to smallest, until the trace norm difference between $Z$ and $W$ falls under $\epsilon$. □

The only prior work to Theorem D.1 that we are aware of is [18, Lemma 28], which only deals with the identity matrix $W = I$ and is loose by a factor of 2. [18] went on to construct a density function based on the EQUAL function (where $W = I$, c.f. Example C.2) that cannot be approximated by a polynomially-sized standard shallow SPN, up to some exponentially small $\epsilon$. This existence result is then strengthened by [7], who showed that under the HR model, for almost every random tensor $\mathcal{W}$, there exists some $\epsilon$ (potentially depending on $\mathcal{W}$), such that any polynomially-sized standard shallow SPN cannot approximate $\mathcal{W}$ under $\epsilon$. Based on Theorem D.1, we now present a complementary result.

**Theorem D.2.** *Fix any $\epsilon > 0$, and sample each entry of the tensor $\mathcal{W} \in \mathbb{R}^{k_1 \times \cdots \times k_d}$ independently and identically from a standard Gaussian distribution, then with high probability,*

$$\epsilon\text{-rank}^1(\mathcal{W}) \geq O(k^{d/2} - \epsilon k^{d/4}). \tag{46}$$

*Proof.* Consider the reshaped matrix $W \in \mathbb{R}^{k^{d/2} \times k^{d/2}}$ of the tensor $\mathcal{W}$. Clearly, each entry of $W$ is again an iid sample from the standard Gaussian distribution. As shown in [42], the smallest singular value of $W$ is $\Theta(k^{-d/4})$, with high probability. Let $r = \epsilon\text{-rank}^1(\mathcal{W})$, then according to Theorem D.1, we have

$$\epsilon \geq \sum_{j=r+1}^{k^{d/2}} \sigma_i(W) \geq (k^{d/2} - r)k^{-d/4}. \tag{47}$$

Rearranging we obtain the claimed lower bound. $\square$

The failure probability in Theorem D.2 depends on $d$ only mildly: up to a small constant it approaches 0 at the rate $c^{k^{d/2}}$ for some constant $0 < c < 1$. Moreover, the standard Gaussian distribution can be replaced with any subgaussian distribution, or more generally any distribution with a bounded 4-th moment. To see the significance of Theorem D.2, let us note first that we can fix $\epsilon$ beforehand so there is no dependence on the tensor $\mathcal{W}$. Secondly, Theorem D.2 implies that with high probability, for any mixture density $f$, even if we contend with a constant approximation accuracy $\epsilon = O(1)$, a standard shallow SPN $\mathfrak{T}$ would still need $O(k^{d/2})$ many product nodes.

# E   More Related Works

The first attempts at rigorously analyzing the effect of depth in a network was in relation to the computational complexity of boolean circuits. A classical result is due to [43] who showed that for every integer $I$, there are boolean functions computed by a circuit with alternating *AND* and *OR* gates of polynomial size and depth $I$; but if the depth is reduced to $I - 1$, an exponential sized circuit would be required. A similar result was proven later by [37]. Another body of work in similar spirit was by [46; 30] proving that solving the $k$-bit parity task by a circuit of depth 2 requires exponentially many gates. A more recent result is due to [34] proving that bounded-depth boolean circuits cannot distinguish some non-uniform input distributions from the uniform distribution. This work by [34] solved a longstanding conjecture in the field.

Classical results for analyzing the expressiveness of neural networks involved results on universal approximation by [35; 31; 38], and by [32] who studied the networks VC dimension. Although, these early results provided general theoretical insights, they were restricted to shallow networks. Recently, several studies have been undertaken to understand the effect of depth on the expressive capacity of a deep network [41; 36; 45; 40; 39]. Most of these works provide separation results between the class of functions that can be efficiently represented by a deep network and those by shallow networks. However, one major limitation of these works is they consider pathological hand-coded network weights that exhibit these extremal properties by design. It is not evident if these class of networks and the hypothesis function class they encode resemble networks and functions used in practice. Therefore, fundamental questions about the expressive power of depth for neural networks used in practice is still not well understood.

Directly relevant to our contributions in this manuscript are recent works in analysing the effect of depths in *Arithmetic Circuits* [9], *Convolutional Arithmetic Circuits* [7] and particularly in *Sum-Product Networks* [24]. The first theoretical results for depth efficiency of SPNs was by [10]. They

constructed two families of functions - $\mathcal{F}$ which formed a full binary tree - and - $\mathcal{G}$ which consisted of $n$ nodes in every layer with each node being connected to $n-1$ nodes in the previous layer - using an SPN with alternating *sum* and *product* layers. Their results establish that any $n$-dimensional function $f \in \mathcal{F}$ can be computed by an SPN of polynomial size but would require a shallow SPN $\Omega(2^{\sqrt{n}})$ hidden units to exactly represent $f$. They further show that for each $d \geq 4$ there exists a function $g_d \in \mathcal{G}$ that can be computed by an SPN of depth $d$ and size $\mathcal{O}(nd)$ but would require $\Omega(n^d)$ sized shallow SPN to compute. However, this work has several limitations. Firstly, the separation results provided are restricted to depth 3 networks and networks in the family $\mathcal{G}$ and $\mathcal{F}$; it is not clear if a similar separation result holds for intermediate depths. Secondly, as the authors themselves state, it does not comment on any separation results when a deep SPN is only to be *approximated* by a shallow SPN. Thirdly, the specific families of functions $\mathcal{F}$ and $\mathcal{G}$ considered by [10] are not shown to be a relevant and universal hypothesis class that occurs in practice. Lastly, the SPNs considered in this work are not *valid* SPNs i.e. they do not encode a probability density function. Furthermore, the analysis is limited to only discrete variables with indicator leaf functions.

[18] extended the work in [10] by proving that there exist functions that can be efficiently computed by a depth $d$ *valid* SPN but would require super-polynomially size for a depth $d-1$ SPN. In particular, they considered the EQUAL function on an array of Boolean variables $\mathbf{x} = (x_1, x_2, .., x_n)$ defined as follows : let $\mathcal{A} = \{1, 2, 3, .., n/2\}$ and $\mathcal{B} = (n/2 + 1, n/2 + 2, ..., n)$ be the index partition. Then, EQUAL : $\{0, 1\}^n \to \{0, 1\}$ where EQUAL$(\mathbf{x}) = 1$ when $\mathbf{x}_\mathcal{A} = \mathbf{x}_\mathcal{B}$ (i.e. the first half of the input is equal to the second half) and 0 otherwise. They proved that a *valid* shallow SPN would require $2^{\frac{n}{2}}$ units in the hidden layer to exactly represent EQUAL$(\mathbf{x})$ while an SPN of depth 4 would require $\mathcal{O}(n)$ size. Further, they also proved that a shallow SPN would still require $2^{n/2-2}$ nodes in the hidden layer to approximate EQUAL$(\mathbf{x})$[5]. However, [18] also restricted their analysis in the paper to only Boolean variables primarily because they used previous works from circuit complexity on arithmetic circuits to derive their results. Further, for the separation results, they constructed an example restricted to Boolean variables and indicator functions in the leaves; the proof does not generalize to *valid* SPNs with arbitrary density functions. Most importantly, they use very specific hand-crafted functions to prove separation results both for exact and approximate representation with no information on how frequently these functions occur in practice. Therefore, it might be the case that expect for a few hand-crafted pathological example, a shallow SPN can efficiently represent all functions derived from a deep SPN.

Recently, [7] proposed a deep network which they called *convolutional arithmetic circuits* that incorporates locality, sharing and pooling. They went on to show that this network corresponded to the Hierarchical Tucker Tensor decomposition [13]. Their main theoretical result showed that except for a negligible set of measure zero, all functions that can be represented by a deep convolutional arithmetic circuits of polynomial size, require an exponential sized shallow network to be realized exactly or approximated. The hypothesis class they considered was universal. However, a major limitation of their main result is that it is an existence result. That is, they say, for any deep convolutional arithmetic circuit, there exists an $\epsilon$ such that no shallow network of polynomial size can approximate the deep network within this $\epsilon$ distance. However, they do not provide any explicit relation with $\epsilon$ for the approximation. In other words, according to this analysis, this $\epsilon$ which requires an exponentially sized shallow network to approximate a deep network may be infinitesimally small. Therefore, a natural question to ask is : what is the $\epsilon$-dependency of the size of a shallow network approximating a deep network within some $\epsilon$ distance?

# F   Detailed Experiments

We perform experiments on both synthetic and real world data to reinforce our theoretical findings. In appendix F.1, we present experiments on synthetic data to demonstrate the expressive power of an SPN and the algorithm proposed in (7)-(8) which we call SPN-CG. Next, we present two sets of experiments on real world datasets - in appendix F.2, we present results for image classification under missing data. In **??**, we compare the performance of SPN-CG to structure learning techniques for SPNs on seven real world datasets used previously as benchmarks.

## F.1 Synthetic data

In the first experiment, we generated 2000 samples from a 4 component and two dimensional GMM with full covariance matrices for each component. We estimate this GMM using SPN-CG. In each iteration, we add an additional SPN from $\text{TMM}_{2,2}$ and learn its parameters and the mixing weights. We use SGD with weight decay to learn the parameters in each iteration. This experiment helps us to demonstrate that an SPN with univariate leaf distributions (thereby resulting in a mixture model with factored distributions) can estimate a GMM with full covariance matrix. Figure 14 shows the convergence of the model to the true negative log-likelihood on a held-out test set as a function of number of iterations.

Figure 14: (Left) Convergence to true negative log-likelihood using SPN-CG (Right) Surface plots for covariance matrices of the components

## F.2 Image Classification under Missing Data

In this section, we demonstrate the efficacy of generative models like deep mixture models learned using SPN-CG for image classification under missing data. We show that deep mixture models for which marginalization is tractable lend themselves naturally for problems under the missing data regime. We perform experiments on MNIST [15] for digit classification and small NORB [16] for 3D object recognition. We keep the same settings for the experiment as described in [26] i.e. we test on two settings of MAR missing distributions - (i) an i.i.d. mask with a fixed probability of missing each pixel, and (ii) a mask obtained by the union of rectangles of a certain size, each positioned uniformly at random in the image.

Our major aim with these experiments is to test our conditional gradient algorithm for high-dimensional real world settings. Therefore, we directly adapt the experiments as presented in [26]. Specifically, we adapt the code for the proposed HT-TMM for SPN-CG by following the details as given in [26]. In each iteration of our algorithm, we add an SPN structure exactly similar to HT-TMM. Therefore, the first iteration of our algorithm (i.e. SPN-CG1) amounts to a structure similar to HT-TMM while additional iterations increase the network capacity. For each iteration, we train the network by using AdamSGD variant of optimization for parameters with a base learning rate of 0.03 and $\beta_1 = \beta_2 = 0.9$. For each added network structure, we train the model for 22,000 iterations for MNIST and 40,000 for NORB.

We compare our results to the following methods :

1. **Data Imputation Methods** : Data imputation methods are a common technique to handle missing data using discriminative classifiers. The algorithm proceeds by completing missing values via some heuristic and passing the results to a classifier trained on uncorrupted data. In our approach, we use a ConvNet for prediction. We tested on the following data imputation methods in our manuscript :

   - Zero data imputation : completing all missing values with zeros.
   - Mean data imputation : completing all missing values with the mean value over the dataset

Figure 15: Performance of SPN-CG on missing data (a) MNIST data with i.i.d missing pixels (b) MNIST data with rectangles of missing pixels (c) NORB dataset with i.i.d. missing pixels (d) NORB dataset with rectangles of missing pixels

- Generative Stochastic Networks [33]

- Non-linear Independent Components Estimation (NICE) [11]

- Diffusion Probabilistic Models (DPM) [44]

2. **LearnSPN** [24]: We used the original code to learn the structure augmented with the class variable and learn the joint probability distribution using CCCP.

For all the algorithms except LearnSPN, we used the modified version of the code as suggested by [26]. Most of the code can be publicly accessed at : `https://github.com/HUJI-Deep/Generative-ConvACs`. We used the original code as suggested by the authors for LearnSPN. For our algorithm, due to time constraints, we could only perform three iterations for both NORB and MNIST dataset. We present the results for these three iterations denoted in the results as SPN-CG1, SPN-CG2 and, SPN-CG3 in this manuscript (see fig. 15a - fig. 15d). Our implementation for SPN-CG is available at : `https://git.uwaterloo.ca/l4mou/SPN`

The results show that SPN-CG performs well in the regime of missing data for both MNIST and NORB. Furthermore, other generative models including SPN with structure learning perform comparably only when a few pixels are missing but perform very poorly as compared to deep mixture models as larger amounts of data is missing suggesting that the structure of deep mixture models is advantageous. These experiments on MNIST and NORB help us conclude that deep mixture models learned using SPN-CG outperform other methods on image classification with missing pixels. Our results compliment the results in [26] where such experiments with state of the art results were presented.

## Additional References

[30] Miklos Ajtai. $\sum_1^1$-formulae on finite structures. *Annals of pure and applied logic*, 24(1):1–48, 1983.

[31] Andrew R Barron. Universal approximation bounds for superpositions of a sigmoidal function. *IEEE Transactions on Information theory*, 39(3):930–945, 1993.

[32] Peter L Bartlett. The sample complexity of pattern classification with neural networks: the size of the weights is more important than the size of the network. *IEEE transactions on Information Theory*, 44(2):525–536, 1998.

[33] Yoshua Bengio, Eric Laufer, Guillaume Alain, and Jason Yosinski. Deep generative stochastic networks trainable by backprop. In *International Conference on Machine Learning*, pages 226–234, 2014.

[34] Mark Braverman. Poly-logarithmic independence fools bounded-depth boolean circuits. *Communications of the ACM*, 54(4):108–115, 2011.

[35] George Cybenko. Approximation by superpositions of a sigmoidal function. *Mathematics of control, signals and systems*, 2(4):303–314, 1989.

[36] Ronen Eldan and Ohad Shamir. The power of depth for feedforward neural networks. In *Conference on Learning Theory*, pages 907–940, 2016.

[37] Johan Hastad. Almost optimal lower bounds for small depth circuits. In *Proceedings of the eighteenth annual ACM symposium on Theory of computing*, pages 6–20. ACM, 1986.

[38] Kurt Hornik, Maxwell Stinchcombe, and Halbert White. Multilayer feedforward networks are universal approximators. *Neural networks*, 2(5):359–366, 1989.

[39] Holden Lee, Rong Ge, Tengyu Ma, Andrej Risteski, and Sanjeev Arora. On the ability of neural nets to express distributions. In *Conference on Learning Theory*, pages 1271–1296, 2017.

[40] James Martens, Arkadev Chattopadhya, Toni Pitassi, and Richard Zemel. On the representational efficiency of restricted boltzmann machines. In *Advances in Neural Information Processing Systems*, pages 2877–2885, 2013.

[41] Razvan Pascanu, Guido Montufar, and Yoshua Bengio. On the number of response regions of deep feed forward networks with piece-wise linear activations. *arXiv preprint arXiv:1312.6098*, 2013.

[42] Mark Rudelson and Roman Vershynin. The least singular value of a random square matrix is $O(n^{-1/2})$. *C. R. Acad. Sci. Paris, Ser. I*, 345:893–896, 2008.

[43] Michael Sipser. Borel sets and circuit complexity. In *Proceedings of the fifteenth annual ACM symposium on Theory of computing*, pages 61–69. ACM, 1983.

[44] Jascha Sohl-Dickstein, Eric A Weiss, Niru Maheswaranathan, and Surya Ganguli. Deep unsupervised learning using nonequilibrium thermodynamics. *arXiv preprint arXiv:1503.03585*, 2015.

[45] Matus Telgarsky. Benefits of depth in neural networks. *COLT*, 2016.

[46] Andrew Chi-Chih Yao. Separating the polynomial-time hierarchy by oracles. In *Foundations of Computer Science, 1985., 26th Annual Symposium on*, pages 1–10. IEEE, 1985.

[47] Yao-Liang Yu and Dale Schuurmans. Rank/norm regularization with closed-form solutions: Application to subspace clustering. *arXiv preprint arXiv:1202.3772*, 2012.

## Footnotes

[5]We direct the reader to [18] for further details on the exact definition of approximation used and the proof.