[Reviews · NeurIPS 2018]

Reviewer 1



Summary: This paper compares various methods (HMM,TMM,HTF,LTM,SPN) of estimating a density mixture. The authors claim to have a filled a gap in the literature by explicitly relating the expressivity of these compact models to one another. The author(s) in particular characterizes when a shallow density mixture can represent a deep density mixture. concluding an inequivalence between HMMs and SPNs. The author(s) also compare the methods up to approximation, concluding that any density mixture can be approximated by shallow and hence tractable SPNs. The referee recommends a weak acceptance. The summary of compact representations of density mixtures is excellent, and the main results address a gap in the literature.

Reviewer 2



The paper discusses connections between multiple density models within the unifying framework of homogeneous mixture models: tensorial mixtures models [1], hidden Markov models, latent tree models and sum-product networks [2] are discussed. The authors argue that there is a hierarchy among these models by showing that a model lower in the hierarchy can be cast into a model higher in the hierarchy using linear size transformations. Furthermore, the paper gives new theoretical insights in depth efficiency in these models, by establishing a connection between properties of the represented mixture coefficient tensor (e.g. nonnegative rank) and a shallow model. Finally, the paper gives positive and somewhat surprising approximation results using [3]. Strengths: + connections between various models, which so far were somewhat folk wisdom, are illustrated a unifying tensor mixture framework. + depth efficiency is analysed as a simple application of this framework. + approximation results and a greedy algorithm are potentially stimulating for future work. Weaknesses: - experiment section is rather messy and not insightful - some prior work is not adequately addressed Quality: The paper is quite timely in discussing and unifying the above mentioned models using homogeneous tensorial mixture models. In some sense, it compiles the gist of multiple research direction in a meaningful manner. However, in some parts the authors exaggerate the contribution and do not well connect with prior art. For example, while it was perhaps not explicitly mentioned in literature that LTMs are a sub-class of SPNs, this connection is rather obvious, especially in light of well known latent variable semantics in SPNs [4]. I also wonder why SPNs are defined as trees in this paper, when they are usually defined as general DAGs [2]. Furthermore, the strict inclusion claims in Theorem 3.1 are not proven: none of the examples in B.2, B.4, and B.6 rule out the possibility that there exists _some_ polynomial-sized model representing the distribution of the respective "higher" model. Clarity: The paper is rather clear, but the experimental section is quite messy. One has to actually guess what is going on here; experimental setup and comparison methods are not clearly described. The experiments are, however, not the main contribution in this paper, which is rather on the theoretical side. Originality. While parts of the paper in isolation would be incremental, in total the paper is original and provides several new insights. Significance: This paper is clearly of significance and has considerable potential impact. [1] Sharir et al., "Tensorial mixture models", 2018. arXiv:1610.04167v5 [2] Poon and Domingos, "Sum-Product Networks: A New Deep Architecture", UAI 2011. [3] Li and Baron, "Mixture Density Estimation", NIPS 2000. [4] Peharz et al., "On the Latent Variable Interpretation in Sum-Product Networks", PAMI 2017. *** EDIT *** I read the authors' rebuttal. I'm happy with it, as they said that they would adequately rephrase Theorem 3.1. Moreover, I'm glad that the authors aim to clean up the experimental section. A good paper, I actually got a new view on the covered material. Definitely worth an accept.

Reviewer 3



In this paper, the authors formally establish connections between arguably popular, unsupervised learning models, e.g., hidden Markov models (HMM), tensorial mixture models (TMM), latent tree graphical models (LTM). Further, more explicit relationship between expressive power and model parameter (i.e., depth of model) was explored. Finally, an efficient algorithm for approximating "deep models" as a shallow model was described. Empirical validation confirms that this algorithm is indeed useful. In overall, it was very interesting to see probabilistic models from different domains to be formally connected in a single framework of SPNs. The connection itself seems to provide a significant connection and has potential to be a stepping stone for more exciting developments from interchanging ideas of different domains. While I fully understand the page constraints of the paper, I would like to suggest more detailed explanations on the models that are dealt in the paper. Especially, it would help the readers greatly if there is derivation of how each models are "homogeneous", i.e., can be reformulated into (1) with explicit description of corresponding density tensor. Finally, from a curious reader perspective, it would be very interesting to see possibility of extensions to the 'intractable' case. Especially, the authors already mentioned that there is an equivalence between tensor-train and HMMs, while tensor-trains can be generalized to tensor networks and HMMs to general Markov (or graphical) models. Minor comments: - The paper's description of latent tree models were constrained to hidden variables with finite values since it indices of tensor can only be finite. However I would like to point out that there is also an interesting work on using tensor with inifinite indices, called "Function-Train: A continuous analogue of the tensor-train decomposition". I suspect that this would correspond to HMM with continuous-valued hidden variables. - typo in line 263: "the accuracy \epsilon there is exponentially small"